# Inference Optimal VLMs Need Fewer Visual Tokens and More Parameters

**Kevin Y. Li**[*1]    **Sachin Goyal**[*1]    **João D. Semedo**[2]    **J. Zico Kolter**[1]
[1]Carnegie Mellon University, [2]Bosch Center for Artificial Intelligence
{kyl2, sachingo, zkolter}@cs.cmu.edu   joao.semedo@us.bosch.com

## ABSTRACT

Vision Language Models (VLMs) have demonstrated strong capabilities across various visual understanding and reasoning tasks, driven by incorporating image representations into the token inputs of Large Language Models (LLMs). However, their real-world deployment is often constrained by high latency during inference due to the substantial compute required by the LLM to process the large number of input tokens, predominantly arising from the image. To reduce inference costs, one can either downsize the LLM or reduce the number of input tokens needed to represent the image, the latter of which has been the focus of many recent efforts around token compression. However, it is unclear what the optimal trade-off is given a fixed inference budget. We first characterize this optimal trade-off between the number of visual tokens and LLM parameters by establishing scaling laws that capture variations in performance with these two factors. Our results reveal a surprising trend: for visual reasoning tasks, the inference-optimal behavior in VLMs is achieved by using *the largest LLM that fits within the inference budget while minimizing visual token count — often to a single token*. While the token reduction literature has mainly focused on maintaining base model performance by modestly reducing the token count (e.g., $5 - 10\times$), our results indicate that the compute-optimal inference regime requires operating under even higher token compression ratios. Based on these insights, we take the first steps toward designing token compression algorithms tailored for high-compression settings, utilizing prompt-based compression of tokens. Our work underscores the performance and efficiency benefits of operating in low visual token regimes and the importance of developing tailored token reduction algorithms for such conditions.

## 1 INTRODUCTION

Recent advancements in Large Language Models (LLMs) have enabled Vision Language Models (VLMs) to perceive, reason, and respond through both text and image inputs (Liu et al., 2023; Alayrac et al., 2022; Dai et al., 2023). Many VLMs are built on top of pretrained vision encoders, such as CLIP, and pass the patch-based tokens from the visual encoder into the pretrained LLM backbone at a one-to-one ratio for visual context. This results in the LLM processing hundreds of tokens per image, overshadowing those from the user prompt and accounting for most of inference time compute. Consequently, deploying VLMs in real-world applications, particularly on consumer-sided edge devices, e.g., monitoring systems, driving assistants, etc., is often limited by the significant inference cost and resulting latency.

To reduce the inference cost of VLMs, many recent works have focused on decreasing, via merging or pruning, the number of visual tokens passed to the LLM without significant performance degradation (Li et al., 2024c; Shang et al., 2024). Alternatively, inference FLOPs, proportional to the number of parameters and number of tokens processed, can be reduced by using a smaller LLM. Since both the LLM size and number of visual input tokens directly affect the VLM's performance, it becomes unclear what the optimal trade-off between the two is. For example, a 4B LLM processing all visual input tokens results in similar inference costs to a 8B LLM processing half the number of original visual tokens — currently, the ideal choice is unknown.

---

[*]Equal contribution, work partially done at Bosch Research.

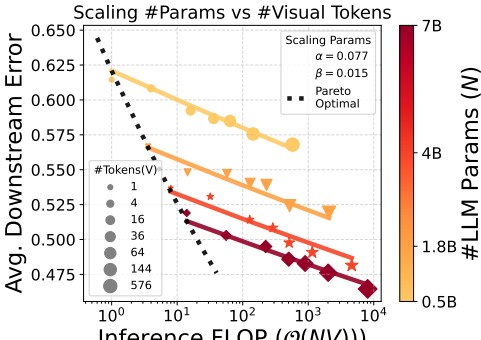 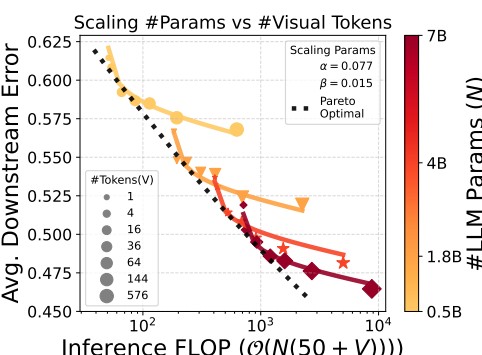

Figure 1: **Inference optimal scaling laws for VLMs.** The number of visual tokens ($V$) passed to the LLM (after token compression, § 2.2), along with the LLM parameter count ($N$), directly determine the inference cost of VLMs ($\mathcal{O}(N(Q + V))$), where $Q$ is the text input tokens. Since the downstream performance of VLMs is directly affected by both these factors, it makes it unclear what the optimal trade-off is for a fixed inference compute. In this work, we try to answer this question with our scaling laws. **Left:** We plot the fitted scaling curves, assuming cached text input tokens (Q=0). We observe a surprising trend: for *visual reasoning tasks*, the compute optimal behavior (dotted black curve) requires using a single visual token with the largest possible language model that can fit under the inference budget. **Right:** Inference optimal behavior under $Q = 50$ requires slightly higher number of visual tokens as the LLM already incurs a fixed cost due to the text tokens.

This observation raises an important question: *given a fixed inference budget, what is the optimal trade-off between LLM size and the number of visual tokens processed for downstream performance?* In this work, we answer this question by building the first inference-time compute-optimal scaling laws for VLMs, modeling performance as a function of both key factors affecting inference cost: LLM size and the number of visual tokens processed. Our scaling laws reveal a striking observation: for visual reasoning tasks, the compute-optimal inference regime entails using the largest feasible LLM with a very small number of visual input tokens — *usually less than 3% the original number of visual tokens*. However, for certain use cases that require detailed image analysis, like Optical Character Recognition (OCR) or document understanding tasks, the optimal approach is quite the opposite, requiring as many visual tokens as possible, as token compression proves ineffective for capturing the dense and diverse information present in such tasks.

Most existing work on token compression has focused on reducing visual tokens by a modest factor (e.g., from 576 to 144 tokens or 64 tokens). In contrast, our results underscore the critical importance of pursuing much higher compression rates (e.g., reducing tokens to 1 or 4) for visual reasoning tasks where such compression is not only feasible, but also compute-optimal. Our work identifies the compute-optimal inference regime for VLMs, emphasizing the importance of high token compression for visual reasoning tasks. We hope these findings will serve as a motivation and foundation for shifting token reduction techniques towards more effective and higher compression ratios.

We first introduce some preliminaries around inference costs and visual token compression for VLMs in Section 2. In Section 3, we formulate and analyze our inference-compute scaling laws, including its generalization across compression techniques and trends across different tasks. Section 4 covers the related work, and we conclude with Section 5.

## 2 PRELIMINARIES

### 2.1 ESTIMATING INFERENCE COST FOR VLMS

The language model in VLMs processes the visual input tokens along with the user text query tokens. As language models become larger, the FLOPs (Floating Point Operations) required to process each input token scales accordingly. We follow the standard practice for estimating the inference time

FLOPs as (Kaplan et al., 2020; Sardana et al., 2024; Snell et al., 2024):

$$FLOPs_{\text{inf}} = \mathcal{O}(N \times T), \tag{1}$$

where $N$ denotes the parameter count of LLM and $T$ denotes the total inference time tokens. We ignore the inference cost stemming from the visual encoder, as we use the same vision encoder with the same input image resolution across all experiments. In addition, many current open-source VLMs currently utilize the same CLIP-L vision encoder (Radford et al., 2021).

We highlight that the *inference cost of VLMs scales proportionally with both the parameters and the number of input tokens processed by the LLM*.

In the context of VLMs, the total inference tokens, $T$, can be further decomposed as $T = Q + V + G$, where $Q$ represents the text input tokens, i.e., the question/prompt, $V$ represents the number of visual tokens from the vision encoder (after token compression), and $G$ accounts for the generated tokens. In many real world applications, such as driving assistance systems, the text input remains constant (e.g., "Alert the driver if the scene ahead has a hazard"). In these scenarios, the text input can be cached, effectively making $Q = 0$ by bypassing self-attention projections and feed-forward calculations. However, in other interactive applications, $Q$ may vary based on dynamic input. We will study the behavior of the downstream error with $FLOPs_{\text{inf}}$ under both the $Q = 0$ and varying $Q$ regimes. In our work, we ignore the $G$ term due to our analyzed tasks' short form responses; however, the analysis with varying $Q$ transfers to $Q + G$ as well.

## 2.2 TOKEN COMPRESSION IN VLMS

As discussed in the previous section, inference FLOPs for VLMs increase proportionally with the number of visual input tokens (e.g., 576 with CLIP-ViT-L visual encoder). Often, the number of visual tokens dominates the total tokens processed by the language model, especially in applications where the text input can be cached or is on the shorter side. Thus, there has been a growing interest in developing approaches to compress the visual information into a fewer number of tokens.

More formally, let the visual encoder be defined as a function $f(\mathbf{I}) = \mathbf{X}$, where $\mathbf{X} \in \mathbb{R}^{n \times d}$ represents a sequence of $n$ vision embedding tokens produced by the encoder from the input image $\mathbf{I}$. Token compression then learns a vision projector $g_\theta(\mathbf{X}) = \mathbf{Y}$ that maps these embeddings $\mathbf{X}$ to $\mathbf{Y} \in \mathbb{R}^{m \times d}$, a compressed sequence of $m < n$ tokens to be processed by the language model ($n = m$ for standard VLMs without any token compression). We refer the reader to Section 4.1 for a detailed discussion on some of the recent token compression algorithms.

Note that token compression *doesn't* refer to using a smaller visual encoder or using smaller image resolutions as inputs to the encoder. These approaches usually either do not decrease the visual token count much (beyond around 224) or lead to a large drops in performance (Li et al., 2024a).

## 3 TOKENS VS PARAMETERS: INFERENCE TIME SCALING LAWS FOR VLMS

The deployment of vision language models in real-world applications comes with significant challenges, particularly surrounding inference latency and frames per second (FPS). For instance, in real-time systems, such as automotive driver assistance or hazard monitoring, maintaining high FPS and quick response times is crucial for safe and effective deployment. Consequently, reducing inference FLOPs while minimizing downstream performance degradation is of critical, practical importance, especially on consumer-grade edge devices, which are often severely compute constrained.

This has led to a growing interest in visual token compression for VLMs (§ 2.2). Alternatively, one could also use a smaller LLM to reduce inference cost. However, both of the above factors directly influence the downstream performance (§ 2.1). This raises a key question: *Given a fixed inference compute budget for VLMs, what is the optimal trade-off between the language model size and the number of visual tokens processed?* In our work, we answer this question by developing scaling laws for VLMs that account for the varying parameter count of the language model component and the number of visual input tokens processed by the language model. As mentioned in Section 2.1, we assume the inference cost from the visual encoder to be fixed and ignore it from here on out.

## 3.1 Tokens vs Parameters: Scaling Law Formulation

Recall that the performance of a VLM is primarily governed by the parameter count of the language model and the number of visual tokens processed by the LLM, assuming a fixed visual encoder. Accordingly, we model the scaling behavior of VLM performance as:

$$Y(N,T) = \frac{A}{N^\alpha} * \frac{B}{T^\beta} + D, \tag{2}$$

where $N$ denotes the LLM parameters, $T$ denotes the total inference tokens, $\{A, B, D, \alpha, \beta\}$ are learnable parameters, and $Y(N,T)$ is a measure of model quality. Although traditional scaling laws have been studied in the context of training loss Kaplan et al. (2020), practitioners often use the direct downstream performance to assess model quality (Gadre et al., 2024; Goyal et al., 2024b; Liu et al., 2022). We use average performance error on a suite of nine commonly used visual reasoning benchmarks (§ 3.2) as a measure of model quality $Y(N,T)$.

Below, we summarize the role of each of these learnable parameter in the scaling law (Eq. 2).

**LLM Quality Parameter ($\alpha$):** This parameter dictates how the downstream error changes with the complexity of the LLM, i.e., its parameter count. A higher $\alpha$ indicates a better language model, such as Llama3-7B outperforming Llama2-7B, often due to superior pretraining.

**Visual Token Quality Parameter ($\beta$):** $\beta$ captures the quality of the visual input tokens fed into the LLM, reflecting the quality of the compression technique. A better token compression algorithm would yield a higher $\beta$, allowing for more reductions of $T$ visual tokens than less effective methods while maintaining the same downstream performance.

**Constants $A, B, D$:** $A$ and $B$ are normalizing constants and $D$ refers to irreducible loss, which cannot be reduced even with the largest $N$-sized language model or all $T$ visual tokens (capped at 576 for our choice of vision encoder).

## 3.2 Experimental Setup

**VLM Training and Evaluation:** We use the LLaVA-Next framework (Liu et al., 2024b) to train VLMs with the Qwen-1.5 family of language models as the backbone. Specifically, we utilize the $\{0.5, 1.8, 4, 7, 14\}$B-chat models (Bai et al., 2023). The pretraining and finetuning dataset and hyperparameters follow Liu et al. (2024a), except we doubled the effective batch size for finetuning.

To estimate the downstream error $Y(N,C)$, we test our trained VLMs on a suite of nine commonly used benchmarks for evaluating visual reasoning: MME (Fu et al., 2024), GQA (Hudson & Manning, 2019), AI2D (Kembhavi et al., 2016), MMBench (Liu et al., 2024c), MMMU (Yue et al., 2023), ScienceQA (Lu et al., 2022), MathVista (Lu et al., 2024), POPE (Li et al., 2023c), and ChartQA (Masry et al., 2022). We average the normalized evaluation metric errors to compute $P(N,C)$. For MME, the Cognition and Perception scores were added and normalized, while the F1 score was used for POPE (Liu et al., 2024a). As previously mentioned in Section 2.1, this set of datasets was selected due to their similar prompt and generation length and overall comprehensiveness.

**Visual Token Compression:** CLIP ViT-L/14 (Radford et al., 2021) is used as the vision encoder for all experiments, and we compress the original 576 tokens to $\{144, 64, 36, 16, 4, 1\}$ using one of our variants of TokenPacker (Li et al., 2024c) which replaces interpolation with a convolution for downsampling (no adding query embedding, refer to App. B for more details).

**Fitting Scaling Laws:** We fit the proposed scaling law (Eq. 2) on $\{Y(N,T), N, T\}$ pairs, with $N \in \{0.5, 1.8, 4, 7\}B$ and $T \in \{1, 4, 16, 36, 64, 144, 576\}$. We use grid-search, for its stability (Goyal et al., 2024b), to estimate the scaling parameters $\alpha, \beta, A, B,$ and $D$. The final scaling law is evaluated on a $N = 14B$ VLM model at various $T$ visual tokens. Further details about the grid-search fit can be found in Appendix A.2.

## 3.3 Results: Estimated Scaling Curves

Figure 1 presents the fitted scaling curves, illustrating the variation in average downstream error as a function of inference FLOPs. The scatter sizes represent the number of visual input tokens

processed by the language model, while the color scale indicates the varying number of language model parameters. We make some key observations below.

**Log-Linear Relation between Error and Number of Visual Input Tokens:** Consider the change in performance for the 7B model as the number of visual input tokens varies (maroon curve in Figure 1.) Recent works on visual token compression (Li et al., 2024c; Shang et al., 2024) claim little to no performance degradation with token compression. For example, they report similar performance to the base model's 576 tokens even when visual token count is reduced to 36 or 144 on certain tasks. However, our scaling curves in Figure 1 reveal a different trend, showing a *log-linear decrease in visual reasoning performance as the number of visual input tokens is reduced*. We believe this discrepancy arises because of the limited downstream benchmarks evaluated in previous works which may not fully capture the VLM's overall capabilities.

**Error Varies $5\times$ Faster with LLM Parameters than with Tokens:** Recall from the scaling law (Eq. 2) that $\alpha$ represents the LLM quality parameter and $\beta$ represents the visual token quality parameter, both denoting the rate at which they influence the downstream error respectively. From Figure 1, we observe that for our selection of language model family (Qwen-1.5) and token compression algorithm, $\alpha = 0.077$ is more than five times larger than $\beta = 0.015$, signifying that VLM error increases significantly faster when reducing the LLM parameters compared to reducing the number of visual tokens. Therefore, when minimizing inference FLOPs, it is more effective to prioritize reducing visual tokens ($V$) first as the impact on performance is less pronounced than reducing the LLM parameters ($N$).

**Scaling Laws Hold for Increases in LLM Scale:** We evaluate the accuracy of our scaling laws (fitted on VLMs of 0.5B-7B range) for predicting the performance for larger models. We estimate the performance of Qwen-1.5 14B using our fitted scaling laws. Our scaling laws estimate the performance with an error margin of less than 2%, as visualized in Figure 2 and Figure 6b. The log-linear relationship between the error and number of visual tokens persists, and the greater influence of the LLM's size compared to visual tokens on performance continues to hold. Thus, for VLMs using 7B language model backbones, it is still optimal to increase LLM size to 14B while reducing visual token count for fixed inference costs.

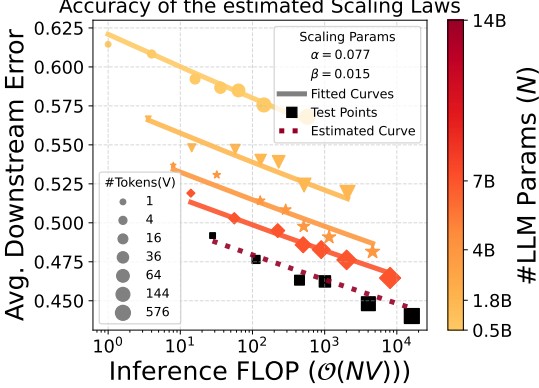

Figure 2: Our scaling laws (fitted on 0.5-7B VLMs), estimate the performance of 14B VLM with an error margin of less than 2%.

### 3.3.1 COMPUTE-OPTIMAL VISUAL REASONING INFERENCE FAVORS MORE LLM PARAMETERS

Observe the pareto optimal curve (black dotted curve) in Figure 1. Our results reveal a striking insight: under a fixed inference budget, visual reasoning tasks heavily favor increasing LLM parameter count. At any given inference FLOPs (denoted by any potential vertical line in Figure 1), the compute-optimal strategy is to allocate FLOPs towards a larger LLM by reducing the number of visual tokens (even to one if the prompt can be pre-computed and cached, $Q = 0$).

A similar trend holds in the $Q = 50$ regime, where the optimal number of visual tokens is around 16, a 97% reduction from the original number of tokens. This can be intuitively explained by the fact that since the language model is already dedicating a fixed amount of compute to process text input tokens, the initial increase in the number of visual tokens represents only a minor amount of the fraction of compute already being spent. Thus, the optimal tokens increases to 16, compared to just 1 in the cached regime.

In Figure 4, we compare VLMs with varying combinations of LLM size and visual token counts under a fixed inference budget. We observe that for many visual reasoning tasks, increasing the size of

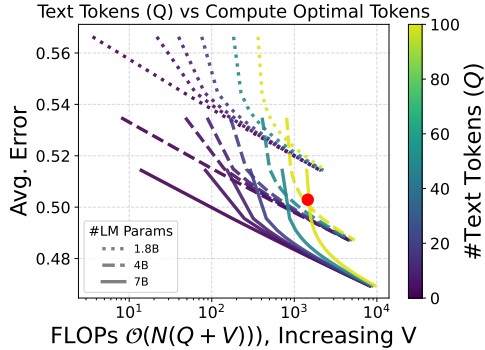 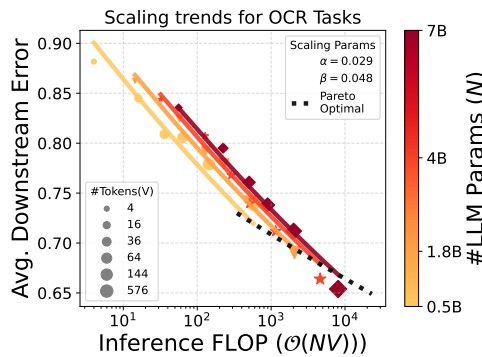

(a) Performance trend changes based on LLM sizes when varying number of input text token $Q$.

(b) Scaling law for VLM token compression and LLM model size on OCR-like tasks.

Figure 3: **Performance trends when shifting input text token count and benchmark family. Left:** For visual reasoning tasks, as the number of text tokens increases, the impact of increasing the number of visual tokens $V$, i.e., reducing compression, becomes more apparent. Intuitively, at a large enough amount of text tokens, initial increases in visual tokens are only a minor fraction of the overall compute. **Right:** When the family of tasks shifts from visual reasoning to OCR/text-understanding, the trends shift: visual token count should be the prioritized instead of LLM size.

the language model while reducing visual tokens can lead to significant relative gains. This may be in part due to the scaling properties of the LLMs themselves, leading to models with stronger world views that can better extrapolate with less visual information than their smaller counterparts (Radford et al., 2021; Wei et al., 2022). We note that this trade-off, while effective for visual reasoning, does not extend to certain tasks, e.g., document comprehension, text identification, etc., where a limited number of tokens may fail to capture the high density of information. We discuss this further in Section 3.4.

**Scaling Inference Compute by Simply Repeating Tokens:** Many recent works around scaling test-time compute by introducing special tokens (Goyal et al., 2024a) or multiple parallel generations (Zelikman et al., 2024) have shown promising gains in reasoning tasks for language models. We test this notion with VLMs by repeating the visual input tokens (compressed to 4) multiple times. However, we do not observe any performance gains. This is most likely due to the downstream tasks for VLMs being not as reasoning-intensive, thus highlighting the importance of developing better token compression algorithms and potentially introducing more challenging benchmarks.

### 3.3.2 Variation in Optimal Tokens with Text Query Length

In the previous section, we observed that when the text input can be cached ($Q = 0$), compute optimal inference requires the use of a single visual token paired with the largest possible LLM that fits under the inference budget. This scenario covers many practical applications, such as monitoring systems or scenarios where text input remains static. However, in interactive systems where the text input can be dynamic and long, i.e., high $Q$, the situation changes.

In Figure 3a, we plot the average downstream error against FLOPs across different lengths of text input tokens ($Q$), with the color of the lines representing the variations in $Q$. When comparing the performance of the 7B model (solid curves) with the 4B model (dashed curves) at a high $Q$ (indicated by the green curves for each model), we observe that there is a sharp increase in error as inference FLOPs are reduced for the 7B model, particularly when visual tokens are reduced significantly. At a certain point (marked by the red dot in Fig. 3a), it becomes more advantageous to use the 4B model with a higher number of visual tokens rather than the 7B model with fewer tokens.

This phenomenon can be understood intuitively: as the LLM processes longer text sequences, the computational cost incurred by text tokens is already considerable. Consequently, increasing the number of visual tokens has a comparatively smaller impact on the overall inference FLOPs. There-

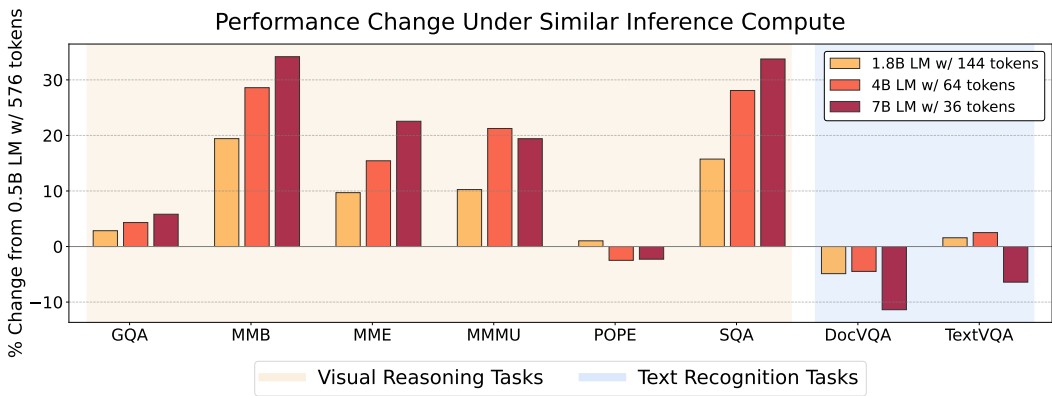

Figure 4: **Performances of various LLM size and visual token count combinations with similar inference compute on two families of tasks.** For many visual reasoning tasks, increasing the LLM size by decreasing the number of visual tokens improves performance. However, for text recognition tasks, decreasing the number of visual tokens is detrimental to performance.

fore, for higher text token lengths ($Q$), increasing the number of visual tokens leads to better performance without significantly increasing the computational burden. Thus, the optimal number of visual input tokens rises with an increase in $Q$. This case demonstrates the need for careful balancing of visual token count and LLM size, especially in scenarios where text inputs are long, to achieve compute-optimal performance without sacrificing accuracy.

### 3.4 Scaling Laws for OCR Tasks

Until now, we have focused on scaling behavior for visual reasoning tasks, highlighting the key finding that using a single visual token with the maximum possible LLM parameters is the inference-optimal configuration. However, is the same valid for all tasks? VLMs have recently been applied to document reading and OCR-style tasks where a single visual token may be insufficient due to the high density of information. Unlike visual reasoning tasks, these tasks lack visual structure in the image and intuitively need more tokens to record the (generally textual) details in the image. We verify the same by fitting our scaling laws (Eq. 2) on DocVQA (Mathew et al., 2021) and TextVQA (Singh et al., 2019) benchmarks, where the tasks require mainly OCR capabilities.

Figure 3b presents the fitted scaling law for OCR tasks. Notably, there are no significant gains in average downstream performance from increasing LLM parameters; instead, the number of visual tokens predominantly dictates the performance. This observation is reflected in the scaling law parameters, where the LLM-quality parameter $\alpha = 0.029$ is nearly twice as smaller than the token quality parameter $\beta = 0.048$. This trend is in stark contrast to the scaling parameters observed for visual reasoning tasks where the LLM-quality parameter ($\alpha$) was more than five times larger than the token parameter (§3.3). This notion of visual tokens playing the significant role in text-in-image recognition and understanding is further echoed in Figure 4, which shows token compression weakens VLM performance despite increasing the capabilities of the LLM component to compensate.

We find that our scaling laws generalize with other visual token compression algorithms. We share additional scaling law results on LLaVA-PruMerge (Shang et al., 2024) in Appendix 3.5.

### 3.5 Generalizing Scaling Laws to other Token Compression Algorithms

We find that the takeways for our proposed scaling laws generalize across visual token compression algorithms. We fit scaling laws with VLMs utilizing LLaVa-PruMerge (Shang et al., 2024), one of the first visual token compression projectors, on $N \in \{0.5, 1.8, 4\}B$, and $T \in \{36, 64, 144, 192, 228, 576\}$ following Section 3.2. Unlike many current projectors (Cai et al., 2024; Hu et al., 2024; Li et al., 2024c), LLaVA-Prumerge suffers massive performance drops in extreme token compression regimes, resulting from its training-free methodology. Therefore, we do not consider these conditions in its scaling laws.

When using the same $A, B, D$ values fit in Section 3.3, we find comparable $\alpha = 0.069, \beta = 0.008$ compared to before ($\alpha = 0.077, \beta = 0.015$, § 3.3). Similar values for $\alpha$ show that our scaling law is capable of capturing the quality of the LLM across VLM architectures and the decrease in $\beta$ shows that the PruMerge compression algorithm "weaker" than TokenPacker, which was also empirically shown in our Table 1. Fitting the scaling laws from scratch results in $\alpha = 0.077, \beta = 0.041$ where performance error only varies around $2\times$ faster with LLM parameters than with tokens. Thus, even across different VLM architectures, compute-optimal inference for visual reasoning and understanding tasks continues to strongly favor the LLM parameter count, as shown in Figure 5.

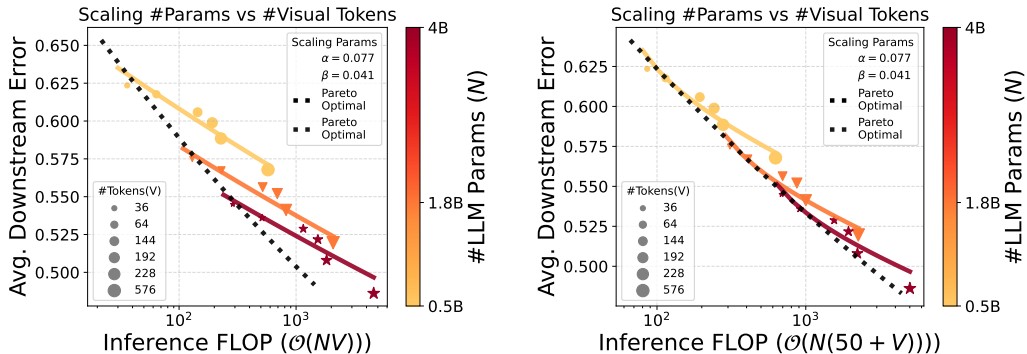

Figure 5: **Inference optimal scaling laws for PruMerge:** When replacing the token compression algorithm, the main findings still hold: inference-optimal behavior is still to increase the LLM parameter count by reducing visual tokens in fixed compute scenarios.

### 3.6 ENHANCING EXTREME TOKEN COMPRESSION ALGORITHMS WITH USER QUERY

For visual reasoning tasks, our scaling laws empirically show that only very few tokens are required for visual reasoning tasks. This need for extreme compression — down to 1, 4, or 16 tokens — to operate compute-optimally represents a paradigm shift compared to existing compression work which focus on moderate compression (e.g., reduction by 75%). We find that utilizing the user's query prompt during the token compression process can be critical for retaining relevant information and minimizing performance reductions when tokens are reduced to as few as 1 or 4.

We build over existing token compression algorithms (Li et al., 2024c), to incorporate query-based token compression. A full summary of the updated algorithm can be found in Appendix B. Following the training regime of LLaVA-1.5 (Liu et al., 2024a), we use the pretrained Vicuna 7B model as the LLM component (Zheng et al., 2023). Based on the importance of high token compression underscored by our scaling laws (§ 3.3), we focus on visual token budgets of $\{1, 4, 16, 36, 64\}$, resulting in compression rates of 88.9% to 99.8%. We benchmark our method on a diverse, comprehensive set of datasets consisting of visual reasoning and OCR/text-understanding tasks: GQA (Hudson & Manning, 2019), MMBench (MMB) (Liu et al., 2024c), MME (Fu et al., 2024), POPE (Li et al., 2023c), ScienceQA (SQA) (Lu et al., 2022), TextVQA (Singh et al., 2019) VizWiz (Gurari et al., 2018), and VQAv2 (Goyal et al., 2017).

We find that incorporating the user's query allows the model to perform better than the baseline and alternative compression algorithms on multiple different datasets. At the one-token level, our variant improves upon the existing algorithm on *all* benchmarks and can mitigate the original TokenPacker's performance gap compared to vanilla LLaVA-1.5 by 34% on VizWiz and ~12% on both POPE and MMBench. The trend continues at the four-token level. Table 1 displays the full results.

## 4 RELATED WORK

### 4.1 TOKEN REDUCTION IN VISION-LANGUAGE MODELS (VLMS)

VLMs are composed of three key components: (a) a visual encoder that encodes the input images, (b) a language model (LM) that processes the visual tokens from the encoder along with the user

text query, and (c) a projector that maps the visual tokens to the input embedding space of the LM. Section A.1 contains additional details exploring various projector designs. Often, the number of visual tokens (576 tokens per image for CLIP-ViT-L, for instance) significantly exceeds the number of text tokens, leading to high inference costs. This disproportionate scaling of visual tokens also hinders multi-frame integration due to the limited context length of the model. Inference cost is a critical factor in many real world applications of computer vision systems. Thus, reducing the number of visual tokens processed by the language model has become an active area of research.

LLaVA-PruMerge (Shang et al., 2024) and Yu et al. (2024) propose training-free methods that filter out visual tokens (from CLIP) that have a low similarity with the CLS token. TokenPacker (Li et al., 2024c), on the other hand, learns a compact token compression module using cross-attention over visual tokens, allowing for reduced number of tokens while preserving salient information. While the above approaches focus on token reduction without directly changing the visual encoder (CLIP) output, recent works based on Matryoshka Representation (Cai et al., 2024; Hu et al., 2024) modify the CLIP output directly to generate nested CLIP embeddings for a flexible token count. Zhang et al. (2024) investigate methods that emphasize task-relevant pixels during image processing.

Another approach to reducing inference cost is adaptive token processing, where the compute dedicated to certain tokens at inference is varied Jain et al. (2024). Many methods prune visual tokens within the LLM due to their lower attention scores compared to the prompt, system, etc., tokens (Chen et al., 2024; Wan et al., 2024), a heuristic commonly found in regular text-only LLM KV cache reduction techniques (Zhang et al., 2023; Oren et al., 2024). Finally, while we focus our paper on image-based VLMs, a host of works (Xu et al., 2024; Shen et al., 2024) discuss token compression for video processing using VLMs. We defer a discussion of these to Section A.1.

## 4.2 SCALING LAWS

Understanding how the performance of modern deep networks shifts as key design factors, such as the number of parameters or training tokens, are scaled has become a focal point of research, particularly as these models continue to grow in size and complexity. Scaling laws offer crucial guidance for optimizing the architecture of such models. Notably, Kaplan et al. (2020); Hernandez et al. (2021); Hoffmann et al. (2022) do a thorough investigation into training compute-optimal language models, highlighting the need to scale pretraining tokens and parameters at the same rate. Cherti et al. (2023); Gadre et al. (2023) perform a similar study on scaling laws for CLIP (Radford et al., 2021), corroborating that performance improvements arise from increasing both parameter counts and pretraining image-caption pairs.

Closest to our work, Li et al. (2024a) investigate what factors improve the performance of LLaVA (Liu et al., 2023). They observe performance gains with increasing language model size, visual encoder size, and input resolution. They investigate each of these factors when scaled independently. In contrast, in this work we focus on understanding the optimal trade-off between language model size and the number of visual input tokens, given a fixed inference budget to fit in. Note that in our work, visual input token count is varied (decreased) using token compression algorithms (§ 4.1) and *not* by varying the input image resolution or using a different CLIP model.

While scaling the pretraining of LLMs has led to emergent capabilities, there has recently been a growing interest in improving their reasoning capabilities by scaling inference time compute. Brown et al. (2024) show impressive performance boosts if the language model is allowed multiple attempts on a problem. In fact, Snell et al. (2024) show that scaling test time compute by parallel multiple generations at inference gives performance comparable to a $14\times$ larger model on math tasks. Goyal et al. (2024a) show performance gains by appending special tokens at the end of input to scale test time compute. In contrast, we characterize the optimal trade-off between tokens and parameters, for getting the best performance at a given fixed test time (inference) compute.

## 5 DISCUSSION AND CONCLUSION

In our work, we demonstrate that the optimal trade-off for VLMs inference is to use *very few* visual input tokens along with the largest possible LLM that fits within the budget. This result has quite important consequences. Existing works aim towards moderate reduction in token count (e.g., from 576 to 144), while trying to match the performance of the base model (no token reduction). However,

our results show that the community needs to focus towards extreme token reduction (e.g., down to 1, 4 or 16 tokens), as the inference optimal regime requires very few visual input tokens. Note that although extreme token reduction can lead to a drop in performance compared to the base model, it is still better than using more tokens with a smaller LLM. The performance with very few visual tokens is poised to only improve further as we develop token reduction algorithms tailored for extreme reduction. While our findings are focused on visual token compression at the projector level prior to passing into the LLM, we leave the compute-optimal scaling properties of adaptive token processing algorithms that operate within the LLM component for subsequent work. We hope that these critical insights from our paper will guide future research towards developing better token reduction techniques and thus inference optimal VLMs.

## 6 ACKNOWLEDGEMENTS

We thank Leslie Berberian, Devin Willmott, Qiu Chen, and Vijay Sadashivaiah at the Bosch Center for AI for useful discussions and help with running some of the experiments on Bosch's compute. We also thank Albert Gu for his feedback on the draft. KL and SG are supported by funding from the Bosch Center for Artificial Intelligence.

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

# A APPENDIX

## A.1 ADDITIONAL RELATED WORKS

### A.1.1 VISION PROJECTOR DESIGN

To bridge the gap between the separate image and text modalities presented by the vision encoder and language model respectively, vision projectors map the image tokens from the vision encoder into the language space. Many design choices for the projector exist. Numerous VLMs utilize query-based projectors, which combine the embeddings of visual tokens with that of query tokens via cross-attention or similar mechanisms, like the Q-Former projector introduced BLIP-2 (Li et al., 2023a) and used in following work (Dai et al., 2023; Zhu et al., 2023). Other VLMs use simple linear projectors or MLPs to connect the encoder and LLM (Liu et al., 2023; 2024a; Su et al., 2023). While most architectures use the projectors to create new tokens to feed into the LLM alongside text, some architectures like Flamingo (Alayrac et al., 2022) or CogVLM (Wang et al., 2024a) directly interweave the visual information into the language model. In our work, we will be focusing on projectors that fall in the former category.

### A.1.2 ADDITIONAL APPROACHES FOR EFFICIENT VLMS

Apart from reducing the number of visual input tokens to the language model, people have explored various other techniques, including a mix of quantization (Liu et al., 2024a) and smaller encoders or language models (Yao et al., 2024; Chu et al., 2023; Zhou et al., 2024) for improving inference.

VLMs utilized in video processing often combine decreases in vision encoder output size with token compression techniques to prevent excessive latency and memory constraints. Visual tokens are often merged temporally across frames (Xu et al., 2024; Shen et al., 2024) as well as spatially for individual frames (Xu et al., 2024). Vision encoders, such as Q-Former (Li et al., 2023a), are preferred over more traditional CLIP models due to their ability to extract a smaller fixed number of tokens per image (Weng et al., 2024; Li et al., 2024b). Although compression techniques used for video processing often can reduce token counts by large margins, they are rarely evaluated on image datasets, and when they are, compress visual tokens very little or not at all (Li et al., 2023b).

Adaptive token processing, where the compute dedicated to certain tokens during inference is varied Jain et al. (2024), is another approach to reducing the cost of inference. Many methods prune visual tokens within the LLM due to their lower attention scores compared to the prompt, system, etc., tokens (Chen et al., 2024; Wan et al., 2024), a heuristic commonly found in regular text-only LLM KV cache reduction techniques (Zhang et al., 2023; Oren et al., 2024). Finally, while we focus our paper on image-based VLMs, a host of works (Xu et al., 2024; Shen et al., 2024) discuss token compression for video processing using VLMs. We defer a discussion of these to Section A.1.

## A.2 GRID SEARCH DETAILS

While there are many choices of optimizer for fitting the scaling laws like curve-fitting in SciPy, gradient descent based solvers, etc. We observed that these are not stable and give varying solutions. We converged to using grid-search to fit the scaling laws, similar to the recent works like Goyal et al. (2024b). The grid-search range for each of the parameters were as follows: $\alpha, \beta \in \{0, 0.1\}, A, B, D \in \{0, 1\}$.

## A.3 ADDITIONAL RESULTS FOR SCALING LAWS

We find that our original scaling laws are able to generalize and predict the performance of VLMs at the 14B scale despite only being fitted up to the 7B scale. Our predictions result in less than 2% error between the predicted and actual VLM performance on visual reasoning and understanding tasks at the 14B model parameter scale. Performance is measured as described in Section 3.2.

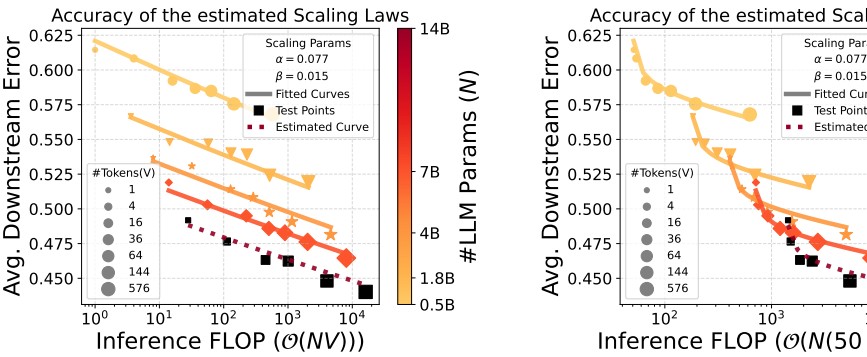

(a) Scaling law prediction for 14B LLM VLM at $Q = 0$.

(b) Scaling law prediction for 14B LLM VLM at $Q = 50$.

Figure 6: **Scaling law predictions at various** $Q$. The scaling laws fitted based on LLMs up to the 7B scale generalize well to the 14B scale, resulting in less than 2% error between predicted and actual VLM performance.

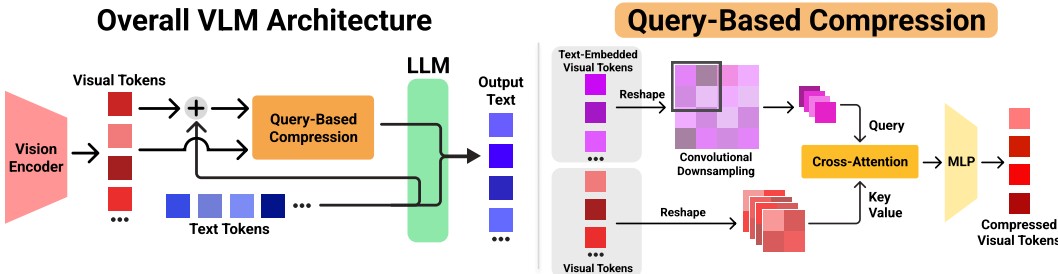

Figure 7: **Our query-based convolutional cross-attention compression technique**. User input text tokens are first processed through the LLM backbone to generate text embeddings that are then combined with the visual tokens. Within the projector, the query-embedded visual tokens are downsampled via convolution. Next, local cross-attention is applied between the downsampled tokens and their respective visual tokens regions. The compressed tokens finally pass through an MLP into the LLM.

## B  PROMPT-BASED TOKEN COMPRESSION

Our scaling laws show that, for visual reasoning tasks, it is optimal to utilize the largest LLM backbone by reducing the number of visual tokens to remain within budget. Many current token compression algorithms focus on compression to 144, 64, or 36 tokens; however, the token count can be as low as 1 for many situations. Thus, improving the ability of extreme compression algorithms would enable the use of larger LLM-backbones for the same inference cost, achieving optimal inference compute usage.

The following section details our updates to the existing token compression algorithm (Li et al., 2024c) to incorporate query-based token compression. Figure 7 summarizes our query-based convolutional cross-attention compression technique.

**User Query Information Injection:**  To make our projector prompt/query-dependent, we add the text embedding of the user's most recent prompt to the image embeddings from vision encoder. We do this by taking the last hidden states prior to the LM head of the user input from the language model as the representation of the user's overall query. The hidden state converted into the text embedding via a linear projection and added to the image visual token embeddings. These fused tokens are later used as the query component for cross-attention. The text-embedding can easily be cached for applications where the prompt is static or is part of a predetermined set. Even if

| Method | # Token | GQA | MMB | MME | POPE | SQA | TextVQA | VizWiz | VQAv2 |
|---|---|---|---|---|---|---|---|---|---|
| LLaVA-1.5 | 576 | 62.0 | 64.3 | 1510.7 | 85.9 | 66.8 | 58.2 | 50.0 | 78.5 |
| PruMerge | ~32 | 57.2* | 60.9 | 1350.3 | 76.3 | 68.5 | **56.0** | 45.2* | 72.0 |
| TokenPacker (TP) | 36 | 59.6 | 62.8 | 1440.9* | 83.3* | 71.0* | 53.2* | 50.2 | 75.0 |
| Matryoshka Multi. | 36 | 60.3 | **64.8** | – | 85.5 | – | – | **52.8** | – |
| Matryoshka Query | 36 | 58.8 | 63.4 | 1416.3 | 81.9 | 66.8 | – | 51.0 | 73.7 |
| **Updated TP (Ours)** | 36 | **60.5** | 62.5 | 1442.0 | 84.5 | 70.6 | 53.3 | 50.1 | **75.8** |
| TokenPacker | 16 | 58.9* | **62.7*** | 1378.8* | **83.7*** | 68.1* | **52.5*** | **50.5*** | 74.4* |
| Matryoshka Query | 16 | 57.6 | 61.9 | **1408.5** | 80.8 | 67.5 | – | 49.8 | 71.1 |
| **Updated TP** | 16 | **59.0** | 62.2 | 1408.0 | 83.4 | **70.7** | 51.3 | 47.7 | **74.5** |
| TokenPacker | 4 | 56.2* | 61.5* | 1347.6* | 81.7* | 68.5* | **49.2*** | 45.7* | 70.5* |
| Matryoshka Query | 4 | 53.0 | 56.5 | 1176.1 | 77.6 | 65.1 | – | **49.4** | 64.1 |
| **Updated TP** | 4 | **56.5** | **62.1** | **1390.3** | **81.8** | **68.6** | 48.7 | 45.0 | **70.6** |
| TokenPacker | 1 | 53.4* | 58.7* | 1262.4* | 80.7* | 69.4* | 46.2* | 41.1* | 66.9* |
| Matryoshka Multi. | 1 | 52.6 | **59.5** | – | 78.4 | – | – | **49.4** | – |
| Matryoshka Query | 2 | 50.8 | 54.4 | 1144.0 | 74.5 | 65.0 | – | 48.5 | 61.0 |
| **Updated TP** | 1 | **53.5** | 59.4 | **1269.1** | **81.3** | 69.9 | **46.8** | 44.1 | **67.3** |
| No Visual Tokens | 0 | 37.7 | 21.0 | 697.8 | 45.4 | 63.6 | 41.7 | 44.4 | 41.0 |

Table 1: **Comparison of various token compression methods for VLMs at different compression rates.** All models use the Vicuna-1.5 7B model as the language backbone. A * denotes benchmark results for other techniques we evaluated, while best scores are **bolded**, and second best underlined. Our updated method outperforms alternatives, including its predecessor, on almost all benchmarks at extremely high compression regions (visual tokens reduced to 1 or 4), highlighting the benefit of utilizing the user query in the compression algorithm for these situations.

the prompt varies, the text-embedding can be pre-calculated prior to processing the image and KV values cached and re-used when processing the visual and text tokens together for generation.

**Token Downsampling with Cross-Attention and Learnable Convolutions:** To compress the number of visual tokens passed into the LLM, we utilize a region-based, cross-attention mechanism that downsamples the vision encoder tokens, $\mathbf{X}$, into a more information-dense form. The mechanism hinges on the property that the $\mathbf{X}$ can be viewed as a $\sqrt{n} \times \sqrt{n}$ grid due to the vision encoder's patchification of the image. Li et al. (2024c;d) passes the "2D" version of $\mathbf{X}$ through a downsampling function that compresses the input by a $s^2$ factor where each resulting token corresponds with a $s \times s$ region in the original input. After this, cross-attention is applied independently between each downsampled token and the corresponding tokens in its $s \times s$ region. We improve upon bilinear interpolation-based downsampling techniques (Li et al., 2024c; Wang et al., 2024b) by using a learnable depth-wise 2D convolution filter of kernel size and stride $s$, providing better expressivity.

### B.1 QUERY-BASED CONVOLUTIONAL CROSS-ATTENTION RESULTS

Table 1 presents the results of our QueCC algorithm in comparison to previous methods, including TokenPacker (Li et al., 2024c), LLaVa-PruMerge (Shang et al., 2024), Matryoshka Multimodal Models (Cai et al., 2024), and Matryoshka Query Transformer (Hu et al., 2024), in low token regimes. We find that at our method performs better than alternatives at the highest levels of compression on multiple different datasets. At the one-token level, our method outperforms other methods on six of the eight datasets and is able to mitigate some of the shortcomings of the original TokenPacker by reducing its gap between vanilla LLaVA-1.5 on VizWiz by 34% and ~12% on both POPE and MMBench. The trend continues at the four-token level. Our model also exhibits strong performance on GQA, MME, SQA, and VQAv2 across compression rates, signaling the prospects of using the user's query to identify key tokens.

### B.2 ABLATIONS OF PROMPT-BASED TOKEN COMPRESSION

We ablate the importance of the query injection and convolutional downsampling components and report the results in Table 2. We find at extreme levels of compression that combining query and convolution can magnify the benefits of either adding only query or only convolution; e.g., TextVQA performance at token count one increased by 0.7 percentage points (pp) with both convolution and

| # Token | Model | GQA | MMB | MME | POPE | SQA | TextVQA | VizWiz | VQAv2 |
|---|---|---|---|---|---|---|---|---|---|
| 1 | Conv and Query | 53.5 | **59.4** | **1269.1** | **81.3** | **69.9** | **46.9** | 44.1 | **67.3** |
|  | Query Only | 53.3 | 59.2 | 1267.7 | **81.3** | 68.8 | 46.3 | 41.7 | 66.6 |
|  | Conv Only | **53.6** | 57.5 | 1215.5 | 80.6 | 69.1 | 46.4 | **45.6** | 66.7 |
|  | No Conv, No Query | 53.4 | 58.7 | 1262.4 | 80.7 | 69.4 | 46.2 | 41.1 | 66.9 |
| 4 | Conv and Query | 56.5 | **62.1** | **1390.3** | 81.8 | 68.6 | 48.7 | 45.0 | **70.6** |
|  | Query Only | 56.4 | 62.0 | 1345.9 | **82.3** | **70.7** | 48.8 | **46.5** | **70.6** |
|  | Conv Only | **56.7** | 60.6 | 1310.4 | 82.1 | 69.0 | **49.4** | 41.3 | 70.5 |
|  | No Conv, No Query | 56.2 | 61.5 | 1347.6 | 81.7 | 68.5 | 49.2 | 45.7 | 70.5 |
| 16 | Conv and Query | **59.0** | 62.2 | **1408.0** | 83.4 | **70.7** | 51.3 | 47.7 | **74.5** |
|  | Query Only | 56.6 | 61.4 | 1354.3 | 82.1 | 69.6 | 50.7 | 41.2 | 71.5 |
|  | Conv Only | 58.9 | 62.5 | 1402.3 | 82.5 | 69.6 | **52.6** | 45.7 | 74.1 |
|  | No Conv, No Query | 58.9 | **62.7** | 1378.8 | **83.7** | 68.1 | 52.5 | **50.5** | 74.4 |

Table 2: **Comparison of model ablations across different token counts and configurations.** Best scores are **bolded**, and second-best scores are underlined for clarity. Adding both query and convolutional components can help boost the baseline performance and can mitigate performance drops that are associated with each individual component.

query while using only one of the components led to at most 0.2 pp increase. In addition, combining the two can mitigate performance drops that are associated with utilizing only query or convolution, as seen in MMB at one token where using only convolution drops performance by more than 1 pp but performance can not only be restored but also improved when adding query, eventually outperforming the baseline by 0.7 pp; a similar situation can be seen for MME.

| # Token | Model | GQA | POPE | SQA | TextVQA |
|---|---|---|---|---|---|
| 16 | LLaMA-VID | 58.2 | 83.1 | 67.4 | 50.8 |
|  | QueCC | 59.0 | 83.4 | 70.7 | 51.3 |
| 4 | LLaMA-VID | 56.2 | 83.5 | 68.7 | 49.1 |
|  | QueCC | 56.5 | 81.8 | 68.6 | 48.7 |

Table 3: **Comparison of LLaMA-VID and QueCC models across different visual/content token counts.** LLaMA-VID results, obtained from (Li et al., 2023b), utilizes the context tokens, resulting in one addition overall token.

We also compare with LLaMA-VID (Li et al., 2023b) which also has strong performance in extreme compression regimes. We compare the performance of our method to its reported performance at similar token compression levels and show that we are able to outperform it in certain tasks despite LLaMA-VID utilizing a stronger vision encoder (Li et al., 2023b). Our analysis shows that both our approach and theirs are competitive, which also validates the key point we wanted to make that query-based compression is necessary under extreme compression. In addition, LLaMA-VID utilizes a separate text decoder model to process the user query, while our method utilizes the existing LLM within the VLM model.

