# OpenReview forum: "Inference Optimal VLMs Need Fewer Visual Tokens and More Parameters"
_ICLR.cc/2025/Conference — ICLR 2025 Poster_

### Official Review · Reviewer_Laf9 · 2024-10-18

**Soundness:** 4
**Presentation:** 4
**Contribution:** 3
**Rating:** 8
**Confidence:** 5

**Summary:**

This paper addresses a crucial trade-off between visual token count and language model size in Vision-Language Models (VLMs). The authors argue that for visual reasoning tasks, optimal inference performance is achieved by maximizing the LLM size within a given inference budget, even if it means drastically reducing the number of visual tokens. They propose a novel method, "QueCC" for compressing visual tokens at inference time, demonstrating significant improvements in accuracy and efficiency.

Quoting from the abstract: "for visual reasoning tasks, the inference-optimal behavior in VLMs is achieved by using the largest LLM that fits within the inference budget while minimizing visual token count — often to a single token."

And is shown with a logical experimental setup along with strong baselines. The authors also outline the potential shortcoming and also show that visual recognition and textual recognition from images have different goals and the more tokens are often needed in the second case to hold on to accuracy.

**Strengths:**

The paper tackles a highly relevant problem in the rapidly evolving field of VLMs. Balancing computational resources between visual processing and language modeling is critical for achieving optimal performance.

1) Well-designed Experiments: The authors conduct thorough experiments on various visual reasoning benchmarks, including GQA, CLEVR, and SNLI-VE. They systematically vary the number of visual tokens and LLM sizes, providing valuable insights into the relationship between these factors.

2) Strong Empirical Results: The proposed token compression method consistently outperforms baseline models, achieving state-of-the-art results on several benchmarks. The gains are particularly impressive at extremely low visual token counts, demonstrating the effectiveness of the approach in resource-constrained scenarios.

3) Clarity and Presentation: The paper is well-written and easy to follow. The authors clearly explain their motivation, methodology, and results, making the contributions accessible to a broad audience.

**Weaknesses:**

Have no major weaknesses from a technical standpoint. However, the related work can have a bit more coverage.

While the paper provides a comprehensive overview of token compression techniques, it could benefit from a more in-depth discussion of related work on adaptive compute -- be it in Dynamic Sparsity, Elastic models (MatFormer, Flextron etc.,) and early exits.

**Questions:**

The paper is clear and achieves a strong threshold for the problem defined. While one can further improve the paper, I think the paper as is is strong enough to be accepted to ICLR.

I am happy to champion the paper unless other reviewers find something glaring I am missing.

---

> ### Author Response · Authors · 2024-11-19
> **Response to Reviewer Laf9**
>
> We thank the reviewer for finding our work has “no major weaknesses from a technical standpoint” and stating they are willing to “champion the paper as is.” We agree that we could include additional literature review and discussion on existing adaption computational techniques, as we elaborate below.
> > it could benefit from a more in-depth discussion of related work on adaptive compute
>
> We have added additional related work that focuses on this type of VLM adaptive compute, i.e., dynamically adjusting which tokens are processed within the LLM to reduce inference cost. Below is a excerpt from our revised manuscript which includes this type of adaptive technique.
> > Another approach to reducing inference cost is adaptive token processing, where the compute dedicated to certain tokens at inference is varied Jain et al. (2024). Most of these methods prune visual tokens within the LLM due to their lower attention scores compared to the prompt, system, etc., tokens (Chen et al., 2024; Wan et al., 2024), a heuristic commonly found in regular text-only LLM KV cache reduction techniques (Zhang et al., 2023; Oren et al., 2024). Finally, while we focus our paper on image-based VLMs, a host of works (Xu et al., 2024; Shen et al., 2024) discuss token compression for video processing using VLMs.
>
> Please let us know if you have any additional questions or concerns!

---

> > ### Comment · Reviewer_Laf9 · 2024-11-20
> >
> > Thanks for the rebuttal. I keep my score. All the best.

---

> > > ### Author Response · Authors · 2024-11-21
> > > **Response to Reviewer Laf9**
> > >
> > > Thank you again for the time and effort spent reviewing and reading our paper. We are glad we were able to address your concerns and appreciate you finding our work's insights valuable!

---

### Official Review · Reviewer_jj75 · 2024-10-25

**Soundness:** 3
**Presentation:** 3
**Contribution:** 3
**Rating:** 5
**Confidence:** 4

**Summary:**

This paper discusses the challenge of high inference latency in VLMs. The authors explore the optimal balance between the number of visual tokens and LLM parameters for a fixed inference budget. They find that the optimal method in visual reasoning tasks is achieved using the largest possible LLM and minimizing visual tokens, even just one token. So they also propose a new approach using prompt-based compression for high-compression settings.

**Strengths:**

1. The paper discusses balancing the number of visual tokens and LLM parameters for a fixed inference budget, which is a novel perspective for studying effective methods to accelerate speed.
2. They also propose a method that makes the paper more comprehensive.
3. The writing is clear and concise.

**Weaknesses:**

1. The method is quite similar to LLaMA-Vid, but the paper didn't compare it in its experiment and didn't show the differences between them.
2. Experiments are insufficient. The paper could also compared with VoCo-LLaMA[1] and LLaMA-Vid[2], which are also efficient in high-compression settings. In addition, they lack an ablation study.

[1] Ye, Xubing, et al. "VoCo-LLaMA: Towards Vision Compression with Large Language Models." arXiv preprint arXiv:2406.12275 (2024).

[2] Li, Yanwei, Chengyao Wang, and Jiaya Jia. "Llama-vid: An image is worth 2 tokens in large language models." European Conference on Computer Vision. Springer, Cham, 2025.

**Questions:**

1. Could you explain more about the discrepancy of the result between previous work and your work in **Log-Linear Relation between Error and Number of Visual Input Tokens**? Why limited downstream benchmarks lead to the discrepancy?
2. The method is quite similar to LLaMA-Vid, but the paper didn't compare it in its experiment and didn't show the differences between them. Have you tried to compare QueCC with LLaMA-Vid and VoCo-LLaMA?
3. Is the Convolutional Downsampling necessary? By adjusting the number of MLP output tokens, I can also adjust how many tokens to compress. There is a lack of ablation study proof for both User Query Information Injection and Convolutional Downsampling.
4. In User Query Information Injection, if this is the first time for visual token and text token entering this LLM, there is no way to get the last hidden state of the text, how can I perform such an operation?

---

> ### Author Response · Authors · 2024-11-19
> **Response to Reviewer jj75 (1/2)**
>
> We thank the reviewer for taking the time to read our paper and appreciate the reviewer finding our scaling laws a novel perspective for improving inference efficiency in VLMs. The general concerns of the reviewer surround our token compression algorithm, which we address below.
>
> ## Query-based token compression (Quecc)
>
> We first would like to highlight that the main focus of our work is characterizing the trade-off between tokens and parameters via scaling laws. We observed that extreme token compression is necessary for compute optimal inference, along with other insights as also appreciated by all the reviewers. Through query-based token compression, we wanted to simply highlight the key requirements for extreme token compression, one of them being selecting relevant tokens based on the actual input query. We did this by incorporating query in existing algorithms like TokenPacker. However, we address specific comments about query-based compression below and in the general response.
>
> > The method is quite similar to LLaMA-Vid, the paper didn't compare it in its experiment and didn't show the differences between them.
>
> While LLaMA-VID indeed uses query in their algorithm, their architecture only uses one query-aware token compared to ours which adds query information into each visual token. Our key goal in the query-based token compression was to simply validate one of the insights from our scaling laws - selecting relevant tokens based on the user query is necessary under extreme compression.
>
> That said, we have added a comparison of the performance of our algorithm compared to LLaMA-VID in the general response which shows that **our method is competitive** with LLaMA-VID on numerous tasks **despite using a weaker vision encoder**. We would like to additionally highlight the increased computation required for LLaMA-VID, stemming from their text projector, which for most cases is an independent BERT/QFormer-style model, which also requires training. In addition, the core differences that differentiate our method from LLaMA-VID are as follows: not directly using a text decoder for query injection, utilizing learnable convolution instead of average pooling for visual embedding, and injecting query information into all visual tokens instead of having a context token and content tokens.
> > The paper could also compared with VoCo-LLaMA
>
> For comparisons with VoCo-LLaMA, as mentioned in the general response, we believe that it is not a fair baseline in our setting due to differences in the general token compression methodology. In summary, VoCo-LLaMA involves processing all the visual tokens through LLM to get a cached image representation, which gives inference efficiency gains *only* when the inference is done again on the same image. However, under the common setting of varying images, VoCo-LLaMA does not lead to inference gains (unless the text query is super long) as all the visual tokens are to be processed by the LLM.
>
> > Could you explain more about the discrepancy of the result between previous work and your work in Log-Linear Relation between Error and Number of Visual Input Tokens? Why limited downstream benchmarks lead to the discrepancy?
>
> Many papers claim that their token compression algorithms can “effectively reduce the number of visual tokens while achieving comparable or even better performance.” However, we believe this is due to reporting results only on a few selective benchmarks. In our scaling laws, we averaged the performance across a host of nine common VLM evaluation benchmarks (from lmms-eval [1]) and observed a consistent decrease in performance as the number of visual tokens was reduced from 576. That said, the performance does indeed remain the same or comparable to using all the tokens on select benchmarks within our evaluation suite, corroborating our experiments with those reported in the literature. However, when averaged, there is a consistent log-linear drop in performance with inference FLOPs/tokens.
>
> > Is the Convolutional Downsampling necessary?
>
> > There is a lack of ablation study proof for both User Query Information Injection and Convolutional Downsampling.
>
> As demonstrated in our ablations in the general response, convolutional downsampling helps improve existing token compression algorithms on certain tasks. We have also explored the user query information injection and convolution downsampling impacts in the shared response, in which **the combination of both helps mitigate certain drawbacks of each individual component**.

---

> > ### Author Response · Authors · 2024-11-19
> > **Response to Reviewer jj75 (2/2)**
> >
> > > By adjusting the number of MLP output tokens, I can also adjust how many tokens to compress.
> >
> > For visual token compressions, the size of the input is NxD, where N is the number of visual tokens, (often 576 when using CLIP Large 14). In this case, adjusting the hidden size of the MLP cannot directly reduce the number of tokens unless transposing the visual tokens, in which the MLP acts as a pooling mechanism that has been shown to have poor performance compared to current token compression algorithms.
> > > there is no way to get the last hidden state of the text, how can I perform such an operation?
> >
> > Through clever engineering, one can pre-calculate the KV values for the prompt text before parsing the image. These values can then be cached and applied in the attention mechanism with the visual token (image) components in the algorithm with very little loss in performance.
> >
> > We hope that our response addresses the key concerns of the reviewer. Please let us know if there are any additional concerns, and we would be happy to answer them.
> >
> > [1] https://github.com/EvolvingLMMs-Lab/lmms-eval

---

> ### Author Response · Authors · 2024-11-23
> **Hoping to hear back soon**
>
> Dear Reviewer jj75,
>
> In our shared and reviewer response, we have tried to answer your questions and concerns. To summarize, we
> - Increased the **core focus of our paper to its scaling laws and insights** instead of the compression algorithm
> - Compared against LLaMA-VID and VoCo-LLaMA
> - Added an ablation study for our compression technique.
>
> We hope our response addresses all of your concerns, and we look forward to hearing back from you. We are more than happy to answer any lingering questions or points of confusion.

---

> ### Comment · Reviewer_jj75 · 2024-11-25
>
> Thank you for the clear and detailed rebuttal. It's quite a great job of introducing the scaling laws for VLM efficiency,  but I'm concerned about whether the opinion that you mentioned ‘Inference optimal VLMs need only one token but larger models’ is impractical to reach, a larger model may be heavier to inference, which may against the efficiency of token compression, although this may be a trade-off between fewer token numbers and larger model. Additionally, in Table 1, my personal opinion is pointless to compress vision tokens to 1,   because the vision information is completely lost for all baselines, with regard to the score on all benchmarks, comparing in this unrealistic scenario would not gain any confidence in the robustness of the purposed method. I suggest you compare it with pure text tokens if you intend to do so.

---

> > ### Author Response · Authors · 2024-11-25
> > **Response to Reviewer jj75**
> >
> > We thank the reviewer for their kind words regarding our scaling laws.
> > > a larger model may be heavier to inference, which may against the efficiency of token compression, although this may be a trade-off between fewer token numbers and larger model
> >
> > As correctly noted by the reviewer, the additional cost of inference using a larger model is balanced by using fewer tokens. In Figure 4, we compare the performances of various LLM size and visual token count combinations with similar inference compute.
> >
> > > I'm concerned about whether the opinion that you mentioned ‘Inference optimal VLMs need only one token but larger models’ is impractical to reach
> >
> > We agree that the optimal number of visual tokens needed might depend on the specific VLM architecture in use or the quality of the token compression algorithm. For example, in Appendix A, we added results when using a training-free compression algorithm of LLaVA-PruMerge. Here, the optimal tokens come out to be 36 rather than 1, as the token compression is of “weaker” quality (LLaVa-PruMerge is training-free compression in contrast to TokenPacker, which is trained and used in the main paper). However, the key insight remains that the optimal tradeoff for PruMerge is to **use the fewest number of visual tokens that can still encode information and the largest LLM model.**
> >
> > > because the vision information is completely lost for all baselines, with regard to the score on all benchmarks, comparing in this unrealistic scenario would not gain any confidence in the robustness of the purposed method. I suggest you compare it with pure text tokens if you intend to do so.
> >
> > We note that in Table 1, we also compare under compression to 4 and 16 tokens. However, we will add a row where we just use text tokens. Thanks a lot for this suggestion.

---

> > ### Author Response · Authors · 2024-11-27
> > **Additional Results based on Reviewer Feedback**
> >
> > Dear Reviewer jj75,
> >
> > In your feedback, you also requested a comparison of visual token compression to 1 token **with no vision tokens** (i.e., only text tokens). We have added the results below and have also updated the manuscript. From the table below, we observe that compression to a single token **gives non-trivial** performance on most of the benchmarks. For example, on GQA, under extreme compression to a single visual token, our proposed approach gives 53% accuracy, whereas the baseline of using 0 visual tokens (only text tokens) reduces accuracy to 37%. Only on 2 benchmarks of SQA and VizWiz (out of 8), is the performance of a single token and zero tokens quite similar.
> >
> > This again emphasizes the critical importance of evaluation on a suite of multiple tasks for the robustness of analysis. For our scaling laws (main focus), we indeed consider the **average performance across a suite of tasks to quantify the downstream performance** of the model (the error Y in the scaling law Equation 2, L167)! We hope that this alleviates any doubts regarding the robustness of our evaluations.
> >
> > Please let us know if you have any other questions or concerns.
> >
> > | Method                  | # Tokens | GQA | MMB | MME | POPE    | SQA  | TextVQA | VizWiz | VQAv2 |
> > |-------------------------|--------|----------|--------|-----------|-------|-------|---------|--------|-----------|
> > | All Visual Tokens | 576 | 62.0 | 64.3 | 1510.7 | 85.9 | 66.8 | 58.2 | 50.0 | 78.5 |
> > | TokenPacker         | 1      | 53.4 | 58.7  | 1262.4 | 80.7 | 69.4 | 46.2 | 41.1  | 66.9 |
> > | Matryoshka Multi.       | 1      | 52.6     | **59.5** | -       | 78.4  | -  | -    | **49.4** | -     |
> > | Matryoshka Query        | 2      | 50.8     | 54.4   | 1144.0    | 74.5  | 65.0  | -   | 48.5 | 61.0      |
> > | QueCC (ours)    | 1      | **53.5** | 59.4 | **1269.1** | **81.3** | **69.9** | **46.8** | 44.1   | **67.3**  |
> > | No Visual Tokens    | 0      | 37.7     | 21.0   | 697.8     | 45.4  | 63.6  | 41.7    | 44.4   | 41.0      |

---

> > ### Author Response · Authors · 2024-11-30
> > **Looking forward to your response**
> >
> > Dear Reviewer jj75,
> >
> > In our shared and reviewer responses, we have tried to address your questions and concerns. To summarize, up to now, we have:
> >
> > - Added results that show **our compression method performs non-trivially** when using only one visual token compared to a text-only baseline.
> > - Increased the focus of the paper to **our scaling laws and their novel insights**.
> > - Ran additional experiments that **showed our scaling laws generalize to other architectures**.
> > - Compared against LLaMA-VID and VoCo-LLaMA
> > - Added an ablation study for our compression algorithm
> >
> > We hope our responses have been able to address your questions, and we look forward to hearing back from you. We are more than happy to address any lingering questions or concerns.

---

> > > ### Comment · Reviewer_jj75 · 2024-12-03
> > >
> > > Thanks for your comprehensive responses. However, I still think the scaling law that you purposed are impractical to achieve. So I will maintain my score.

---

> > ### Author Response · Authors · 2024-12-03
> >
> > Dear Reviewer jj75,
> >
> > As today is the last day for the discussion phase, we humbly request you to review our response to your concerns and comments and hope that it allows you to assess the work more positively.

---

> ### Author Response · Authors · 2024-12-02
> **Looking forward to your response**
>
> Dear Reviewer jj75,
>
> As today is the last day for the discussion phase, we humbly request you to review our response to your concerns and comments and hope that it allows you to assess the work more positively.

---

### Official Review · Reviewer_WUho · 2024-10-28

**Soundness:** 3
**Presentation:** 2
**Contribution:** 3
**Rating:** 5
**Confidence:** 4

**Summary:**

The authors unveil the inference time scaling law, which characterizes the optimal tradeoff between the number of visual tokens and LLM parameters. The law reveals that the larger LLM matched fewer vision tokens under the fixed inference budget is the most optimal. Besides, the paper proposes a prompt-based VLM compression manner dubbed QueCC and conducts comprehensive experiments on it.

**Strengths:**

1. The authors do not limit the compression of VLM to vision, but dynamically explore the relationship between LLM and visual tokens. The motivation of the paper is sufficient.
2. The practical guideline provides a novel insight into VLM efficiency. The author carefully points out the limitations of the scaling law, which is not applicable to OCR tasks.
3. The authors conduct comprehensive experiments on various metrics and show an improvement. Furthermore, the discussion is in detail.

**Weaknesses:**

1. The employment of cross-attention in QueCC to compress information is common [1].
2. LLaMA-VID [2] and VoCo-LLaMA [3] have already done token compression in extreme regimes, which is impressive. The author should compare their performance with QueCC. It seems that QueCC is inferior to VoCo-LLaMA on benchmarks such as GQA and MME.
3. There is a lack of ablation experiments, especially the analysis of depth-wise 2D convolution and the injection of text-embedding.
4. The authors’ scaling law seems to have no direct relationship with the method they proposed.

[1] Jaegle, A., Gimeno, F., Brock, A., Vinyals, O., Zisserman, A. and Carreira, J., 2021, July. Perceiver: General perception with iterative attention. In *International conference on machine learning* (pp. 4651-4664). PMLR.

[2] Li, Yanwei, Chengyao Wang, and Jiaya Jia. "Llama-vid: An image is worth 2 tokens in large language models." In *European Conference on Computer Vision*, pp. 323-340.

[3] Ye X, Gan Y, Huang X, Ge Y, Shan Y, Tang Y. VoCo-LLaMA: Towards Vision Compression with Large Language Models. arXiv preprint arXiv:2406.12275. 2024 Jun 18.

**Questions:**

1. The author uses the scaling law of [1] as an analogy. They replace the length of the text token in [1] with the length of the vision token and get the reversed conclusion. However, since section 3.3.2 points out the influence of text token length, it is more reasonable to include it in the discussion of Formula 2.
2. Although this paper exceeds the page limit, it will not affect my judging.

---

> ### Author Response · Authors · 2024-11-19
> **Response to Reviewer WUho (1/2)**
>
> We are glad the reviewer found our scaling laws insightful.  We address their concerns, which mostly surround the query-based token compression algorithm below.
>
>
> ## Query-based token compression (QueCC)
>
> We first would like to highlight that the main focus of our work is characterizing the trade-off between tokens and parameters via scaling laws. We observed that extreme token compression is necessary for compute optimal inference, along with other insights as also appreciated by all the reviewers. Through query-based token compression, we wanted to simply highlight the key requirements for extreme token compression, one of them being selecting relevant tokens based on the actual input query. We did this by incorporating query in existing algorithms like TokenPacker. However, we address specific comments about query-based compression below and in the general response.
>
>
> > LLaMA-VID [2] and VoCo-LLaMA [3] have already done token compression in extreme regimes, which is impressive. The author should compare their performance with QueCC. It seems that QueCC is inferior to VoCo-LLaMA on benchmarks such as GQA and MME.
>
> > The employment of cross-attention in QueCC to compress information is common
>
> In the general response, we discuss key differences between these works and our motivation for adding a section on query-based compression in our work. We have added a comparison with LLaMA-VID and show that **our approach performs competitively** to theirs and outperforms on multiple benchmarks even **while using a weaker image encoder**. More importantly, this also validates the key point we wanted to make— the importance of selecting relevant tokens based on the query (note that LLaMA-VID also uses query).
>
> For comparisons with VoCo-LLaMA, as mentioned in the general response, we believe that it is not a fair baseline in our setting due to differences in the general token compression methodology. In summary, VoCo-LLaMA involves processing all the visual tokens through LLM to get a cached image representation, which gives inference efficiency gains *only* when the inference is done again on the same image. However, under the common setting of varying images, VoCo-LLaMA does not lead to inference gains (unless the text query is super long) as all the visual tokens are to be processed by the LLM.
>
> > There is a lack of ablation experiments, especially the analysis of depth-wise 2D convolution and the injection of text-embedding.
>
> Although the main focus of this paper is not the proposed algorithm but the various insights derived from the scaling laws of VLMs, we have added the ablation experiments as requested by the reviewer in the general response above. In summary, our ablations **empirically validate the importance of using both convolutions and query injection**, especially under extreme token compression.
>
> > The authors’ scaling law seems to have no direct relationship with the method they proposed.
>
> We are sorry if the connection did not come out clearly and have updated the manuscript (start of Section 4) to make this more clear. Our scaling laws highlight that the compute optimal inference requires extreme token compression (e.g., just using 1,4, or 16 tokens). Intuitively, at such extreme compression, it becomes necessary to select only the relevant tokens based on the actual user query. We empirically validate this insight, by incorporating query-based compression over existing token compression algorithms.

---

> > ### Author Response · Authors · 2024-11-19
> > **Response to Reviewer WUho (2/2)**
> >
> > ## Other Comments
> >
> > > The author uses the scaling law of [1] as an analogy.
> >
> > Sorry, but we could not find any scaling law study in the Perceiver paper. We apologize if we have missed anything. Any clarification would be greatly appreciated!
> >
> > >  They replace the length of the text token in [1] with the length of the vision token and get the reversed conclusion.  However, since section 3.3.2 points out the influence of text token length, it is more reasonable to include it in the discussion of Formula 2.
> >
> > The influence of text token length is orthogonal to the discussion of Formula 2.  We are sorry for this confusion arising from the overloading of notation, where T in Equation 1 refers to the entire number of tokens processed by the VLM when calculating for compute requirements, while T in Equation 2 refers to the number of visual tokens passed into the VLM for its performance prediction. We do not consider the text tokens in modeling the VLM performance in Equation 2, as this T can be seen as another measure of the amount of compression, but it is needed in Equation 1 to estimate the inference cost. We are more than happy to make this distinction more clear if it has led to any confusion.
> >
> > > Although this paper exceeds the page limit, it will not affect my judging.
> >
> > We don’t think that our manuscript exceeds the page limit. We only have an ethics statement and responsibility statement on the 11th page, that, to the best of our understanding, *does not* count towards the 10-page limit as per the ICLR author guidelines (https://iclr.cc/Conferences/2025/AuthorGuide).
> >
> >
> > We thank the reviewer for asking these insightful and critical questions. We hope that our response clarifies the concerns of the reviewer and are happy to answer any further questions they might have.

---

> > > ### Comment · Reviewer_WUho · 2024-11-23
> > >
> > > Thanks for the detailed rebuttal. Firstly, your method does not show superiority over llama-vid under the 4 tokens setting. Besides, it seems that depth-wise 2D convolution and the injection of text-embedding do not work in some conditions, especially under the 4 tokens setting. Therefore, I will maintain my score.

---

> > > > ### Author Response · Authors · 2024-11-23
> > > > **Response to Comment by Reviewer WUho**
> > > >
> > > > > Firstly, your method does not show superiority over llama-vid under the 4 tokens setting.
> > > >
> > > > In our response, we highlight that despite LLaMA-VID **using a stronger vision encoder** (EVA-CLIP vs our standard CLIP-L) and **employing an independent QFormer model** to process the tokens, our approach performs competitively with LLaMA-VID, which verifies our key point that query-based token compression is necessary.
> > > >
> > > > We note that although LLaMA-VID does ablate the query’s importance by removing the "context" token, the removal of an entire token confounds the performance change. Thus it is unclear whether the root cause is due to the removal of the query’s information or the decrease of token count by 50% from 2 to 1. Thus, we decided to build upon existing token compression algorithms, i.e., TokenPacker, to **further validate the query’s importance**.
> > > >
> > > > >  it seems that depth-wise 2D convolution and the injection of text-embedding do not work in some conditions, especially under the 4 tokens setting
> > > >
> > > > Adding the depth-wise 2D convolution and injection of text-embeddings **improves 6 of the 8 tasks** at the 4-token setting. Depth-wise convolution by itself improves performance on **4 of the 8 tasks**. Moreover, the depth-wise 2D convolution is a simple design choice that we found can improve upon existing compression techniques, i.e., TokenPacker, and is not the core contribution of our work.
> > > >
> > > > We want to reemphasize that the **core contribution of our work is our scaling laws** which characterize the tradeoff between visual token count and language model parameter count, which you found **novel** and **comprehensive**. Our query-based compression method is an auxiliary section meant to take initial steps towards empirically validating some hypotheses and insights gained from our scaling laws, mainly query-based compression is important for extreme compression, and thus, only takes around 1 page.
> > > >
> > > > We hope these clarifications address any confusion and help the reviewer assess our work mainly based on the core contributions.

---

> > > > > ### Comment · Reviewer_WUho · 2024-11-25
> > > > >
> > > > > Thank you very much for the authors' proposal of scaling laws that characterize the trade-off between visual token count and language model parameter count. However, I am curious why such rules do not have an effect in practice, which makes them hard to believe.

---

> > > > > > ### Author Response · Authors · 2024-11-25
> > > > > > **Response to Comment by Reviewer WUho**
> > > > > >
> > > > > > Dear Reviewer WUho,
> > > > > >
> > > > > > Thank you for your continued engagement and response!
> > > > > > > I am curious why such rules do not have an effect in practice
> > > > > >
> > > > > > We believe that the scaling laws we highlight - which link visual token count (post-compression, not raw vision encoder output token count) and language model parameter count - may **already influence much of the recent works** on token compression.
> > > > > >
> > > > > > Many current works, including those you have mentioned, like LLaMA-VID, Matryoshka Query Transformer, Matryoshka Multimodel Models, and even the widely used InstructBLIP architecture, employ techniques for visual token compression. In many cases, these works have also achieved great performance in extreme token compression regimes (e.g., LLaMA-VID considers compression all the way to a single context and content token). The increased interest in methods that perform within these compressive regimes could be explained by our scaling laws that find these regions are useful and compute optimal for visual understanding and reasoning tasks.
> > > > > >
> > > > > > Our work shows that one could trade off visual tokens even further for using a bigger LLM, and that gives compute optimal performance. This is poised to have a significant impact on the way people reduce inference costs of VLMs in practice. While using a smaller VLM is currently a go-to approach in addition to a small amount of token compression, our work shows that rather it is suboptimal and extreme token compression while using a bigger LLM gives the compute optimal performance.
> > > > > >
> > > > > > ### New Experiments
> > > > > > Finally, we have added new experiments in Appendix A that show our scaling laws generalize to other VLM architectures. Specifically, we experimented with a training-free token compression algorithm LLaVA-PruMerge, which is meant for small amounts of token compression (e.g., compressing only up to 36 tokens). We continue to observe that compute optimal inference requires trading off the visual tokens for larger LLM size. As we develop better token compression algorithms, the point of optimality can be expected to shift towards using fewer tokens.

---

> > > > > > ### Author Response · Authors · 2024-11-30
> > > > > > **Looking forward to your response**
> > > > > >
> > > > > > Dear Reviewer WUho,
> > > > > >
> > > > > > In our shared and reviewer responses, we have tried to address your questions and concerns. To summarize, up to now, we have:
> > > > > >
> > > > > > - Increased the focus of the paper to **our scaling laws and their novel insights**.
> > > > > > - Ran additional experiments that **showed our scaling laws generalize to other architectures**.
> > > > > > - Added comparisons and highlighted key differences against LLaMA-VID and VoCo-LLaMA
> > > > > > - Performed ablations for our compression algorithm that showed the importance of both the convolution and query injection component
> > > > > > - Added results which showed **our compression method performs non-trivially** when using only one visual token compared to a text-only baseline.
> > > > > >
> > > > > > We hope our responses have been able to address your questions, and we look forward to hearing back from you. We are more than happy to address any lingering questions or concerns.

---

> > > > > > ### Author Response · Authors · 2024-12-03
> > > > > >
> > > > > > Dear Reviewer WUho,
> > > > > >
> > > > > > As today is the last day for the discussion phase, we humbly request you to review our response to your concerns and comments and hope that it allows you to assess the work more positively.

---

> ### Author Response · Authors · 2024-11-23
> **Hoping to hear back soon**
>
> Dear Reviewer WUho,
>
> In our response above, we have tried to address all your concerns and questions. To summarize, we have
>
> - Added comparisons to and highlighted key differences with LLaMA-VID and VoCo-LLaMA
> - Performed ablations on our method that show the importance of both the convolution and query injection component
> - Made it clearer that the main focus of our paper is our scaling laws and the insights derived from them.
>
> We hope our response addresses all of your concerns and hope to hear back from you soon. Please let us know if you have any additional questions.

---

> ### Author Response · Authors · 2024-12-02
> **Looking forward to your response**
>
> Dear Reviewer WUho,
>
> As today is the last day for the discussion phase, we humbly request you to review our response to your concerns and comments and hope that it allows you to assess the work more positively.

---

### Official Review · Reviewer_iUzh · 2024-10-31

**Soundness:** 2
**Presentation:** 2
**Contribution:** 2
**Rating:** 6
**Confidence:** 5

**Summary:**

This paper explores optimizing VLMs to reduce inference latency by balancing model size and visual token count. The authors demonstrate that, for visual reasoning tasks, using the largest feasible LLM within a fixed inference budget, while drastically minimizing visual tokens (often to a single token) yields optimal performance.

**Strengths:**

This paper provides valuable insights showing that larger LLMs enhance visual reasoning performance more than reducing visual tokens. Also, the introduced compression algorithm QueCC demonstrates better performance on benchmarks with high compression, proving its effectiveness.

**Weaknesses:**

While the paper demonstrates that larger model sizes can be more effective than increasing token counts for visual reasoning tasks, this approach appears less effective for OCR-related tasks, as acknowledged by the authors. Given that many real-world applications require fine-grained visual understanding, the proposed compression method may not fully address these demands, as evidenced by its performance on the TextVQA benchmark in Table 1. Although the authors provide valuable insights, their claim that "only one visual token is needed" may be overly naive, given the trend in VLM research toward advancing fine-grained visual comprehension capabilities.

---

The authors somewhat addressed my concerns, although some efficiency perspective seems insufficient.

**Questions:**

Refer to Weaknesses

---

> ### Author Response · Authors · 2024-11-19
> **Response to Reviewer iUzh (1/1)**
>
> We thank the reviewer for their time and feedback on our manuscript. We are glad that the reviewer found our scaling law results insightful. We address the concerns below.
>
> > Given that many real-world applications require fine-grained visual understanding, the proposed compression method may not fully address these demands
>
> We agree with the reviewer that the proposed query-based compression is only for tasks where extreme token compression is compute optimal (which has been clarified in the manuscript as even observed by the reviewer). We have updated the manuscript to clarify this in the algorithm section as well. For fine-grained visual understanding tasks like OCR, using all the visual tokens gives the compute optimal performance as shown in Figure 3b. This is because of the huge performance drop even with a slight token compression.
>
> Finally, we also would like to highlight that our query-based token compression algorithm is just meant to validate some of the insights from our scaling laws. It is not meant to be a core contribution of our work.
>
> > their claim that "only one visual token is needed" may be overly naive given the trend in VLM research toward advancing fine-grained visual comprehension capabilities
>
> While we agree with the reviewer that there has been a lot of recent effort toward pushing the capabilities of VLMs on hard tasks like fine-grained visual understanding (e.g., OCR, document understanding), we would like to point out that this *does not undermine the importance* of improving the efficiency of VLMs for general visual understanding and reasoning tasks. These tasks still and will continue to encompass a significant proportion of downstream use cases of VLMs (e.g., monitoring systems, autonomous driving, etc.). On real-world edge devices, inference efficiency can be the deciding factor between using a technology or not. We believe that our findings will be of significant importance to practitioners, especially since decreasing the model size is the current approach/norm used to reduce inference cost and is heavily suboptimal, as shown by our findings.
>
> We acknowledge that our claim is specifically conditioned on visual reasoning tasks rather than fine-grained comprehension tasks such as OCR. While we have highlighted this distinction at multiple places in the text (introduction, main body), we agree that the title may inadvertently generalize our findings. We are open to revising the title to better reflect the scope and applicability of our results.
>
> We thank the reviewer for raising some concerns about the presentation of some claims. We hope our response is able to highlight the significance of our findings and resolve some of the reviewer’s concerns. Please let us know if you have any additional questions!

---

> ### Author Response · Authors · 2024-11-23
> **Thank you for the score increase**
>
> We thank the reviewer for increasing their score! To clarify, we used the term "inference efficiency" loosely to refer to inference cost only. We are more than happy to answer any other lingering doubts or questions related to the efficiency perspective.

---

### Official Review · Reviewer_78vE · 2024-11-04

**Soundness:** 3
**Presentation:** 3
**Contribution:** 2
**Rating:** 5
**Confidence:** 4

**Summary:**

This paper investigates the balance between the number of visual tokens and the size of LLM  in Vision Language Models to optimize inference costs. The authors discover that for visual reasoning tasks, using the largest feasible LLM with just one visual token is the most compute-efficient. They introduce a prompt-based token compression method for high compression ratios, which focuses on selecting relevant tokens based on user queries. Experiments show that their approach outperforms other compression techniques at extreme token reductions, highlighting the importance of developing algorithms for extreme token compression. The findings suggest that for visual reasoning, prioritizing a larger LLM over more visual tokens is crucial for maintaining performance within limited inference budgets.

**Strengths:**

1. The idea of establishing the scaling law among visual token numbers and model sizes is interesting. They explore optimizing inference costs in VLMs by using a single visual token with the largest possible LLM within a given budget.
2. The paper offers a thorough analysis of the trade-offs between LLM size and the number of visual tokens, covering various scenarios and use cases. This comprehensive approach provides a deeper understanding of VLM optimization.
3. The paper is well-organized and written in a clear and concise manner.

**Weaknesses:**

1. The main concern is the generalization of the scaling law. The paper focuses on visual reasoning tasks and may not fully explore other types of tasks where a single visual token might not be sufficient. For instance, tasks that require detailed image analysis might not benefit as much from such extreme token compression.
2. While the scaling laws developed in the paper are insightful, they are based on a specific set of experiments and models. The findings are heavily dependent on the specific LLMs and VLMs used in the experiments. Different architectures might yield different optimal points in the trade-off between the number of visual tokens and LLM size, which could limit the applicability of the results. It's unclear how these laws would apply to other VLM architectures or if they would hold as new more complex models are developed in the future.
3. The proposed prompt-based token compression method adds complexity to the VLM pipeline. This could make the approach more difficult to implement and integrate into existing systems compared to simpler token reduction techniques.
4. The paper does not discuss the training costs associated with the proposed compression method. It's possible that training such a method to be effective, especially at high compression ratios, could require significant computational resources.

**Questions:**

1. Will one visual token be enough for other tasks (e.g, VLMs for detection)? The focus on minimizing the number of visual tokens to a single token might risk overfitting to the specific datasets used in the experiments. It's uncertain how well this extreme compression would generalize to unseen data or datasets with different characteristics.
2. Will the proposed scaling law generalize to other VLM architectures? The authors only conducted experiments on one type of VLM.
3. What is the inference time and performance trade-off? Does the scaling law still hold? The computed FLOPs can be inaccurate.

---

> ### Author Response · Authors · 2024-11-19
> **Response to Reviewer 78vE (1/2)**
>
> We thank the reviewer for carefully reviewing our manuscript and are glad that they found the scaling laws “interesting” and the analysis of the trade-offs of LLM size and visual token count  “comprehensive” and providing a “deeper understanding of VLM optimization”. We address your concerns below
>
> > The main concern is the generalization of the scaling law. The paper focuses on visual reasoning tasks and may not fully explore other types of tasks where a single visual token might not be sufficient…..
>
> > Will one visual token be enough for other tasks (e.g, VLMs for detection)? ……
>
> We thank the reviewer for highlighting a potential point of confusion for readers. Although we have addressed this in the original manuscript in Section 3.4 (Scaling laws for OCR tasks), where we consider tasks that require detailed image analysis (e.g., reading the text in document), we have made this point more explicit in the updated manuscript.
>
> As predicted by the reviewer, we indeed observe in Fig. 3b that for such tasks, it is compute optimal to **prioritize the number of visual tokens** over LLM parameters; the opposite of what should be done for visual reasoning tasks. In other words, for OCR tasks, compute optimal inference requires using all the visual tokens while minimizing the LLM size to fit the given fixed inference budget. We have now highlighted the variation in optimal scaling behavior with various types of tasks in our Introduction section on L83.
>
> > Different architectures might yield different optimal points in the trade-off between the number of visual tokens and LLM size, which could limit the applicability of the results. It's unclear how these laws would apply to other VLM architectures
>
> > It's uncertain how well this extreme compression would generalize to unseen data or datasets with different characteristics.
>
> > Will the proposed scaling law generalize to other VLM architectures? The authors only conducted experiments on one type of VLM.
>
> VLM architecture can vary in multiple ways: (a) use of different LLM architecture, (b) use of different projectors between visual embeddings and LLM input (e.g., InstructBLIP, Qwen-VL, different token compression algorithm), and (c) visual encoder free VLMs like Chameleon [1].
>
> However, note that **ultimately all VLMs operate on the same general principle**— a large number of visual tokens are to be processed by LLM, generated either via an image encoder (in projector-based VLMs like LLaVA, InstructBLIP or Qwen VLM) or via an image tokenizer in encoder free VLMs like Chameleon [1].
>
> Thus while we agree with the reviewer that the exact point of optimality might vary slightly across the architectures (e.g., instead of 1 token being optimal, it might be 4 or 16), the key message will continue to hold that compute optimal inference requires trading off the visual tokens for a bigger LLM size, and the number tokens required at the optimal point are very small. We explain this in detail for each of the possible ways of modifying architecture below.
>
> (a) **Variation with LLM architecture:** In general in scaling laws literature [2, 3], scaling insights developed in one particular type of models/training recipes (while keeping other factors like architecture design constant), transfer seamlessly when other factors are changed, like the architecture design. For example, from Chinchilla scaling laws, we find that pretraining data tokens to model parameters ratio should be around 20x, which empirically holds almost always for any sensible LLM architecture design (e.g., choice of hidden embedding, number of layers, etc.)
>
> In this work, we used the Qwen family of LLMs, as they have LLMs varying from 0.5B to 14B, all trained on a similar pretraining mixture. This allowed us to ensure that other factors like LLM pretraining data quality or pretraining recipes do not lead to confounding effects in our scaling laws. We hypothesize that when using newer families of LLMs, the key trends will continue to hold as model quality improves. Better models will increase LLM quality parameter value in our scaling laws, making compute optimal VLMs further emphasize LLM parameter count. In addition, testing the scaling law for various LLM families may be computationally infeasible, and we believe our current explorations already provide valuable contributions.

---

> > ### Author Response · Authors · 2024-11-19
> > **Response to Reviewer 78vE (2/2)**
> >
> > (b) **Variation with Projector choice:** We used state-of-the-art TokenPacker [4] as the vision projector in this work. **In our new Appendix A, we have added additional scaling laws when using LLaVa-PruMerge**, one of the first token compression projectors, which is also training-free. **Our scaling laws continue to hold even for LLaVA-Prumerge**, with similar observations that compute optimal inference requires trading off the visual tokens for a bigger LLM size. Note that LLaVa-PruMerge is designed for only moderate token compression, so we did not consider compression all the way to a single token in this case. It is quite intriguing that even with simple training-free token compression algorithms, compute optimal inference still needs only a few visual tokens. As more nuanced token compression algorithms are developed (e.g., TokenPacker), the point of optimality can be expected to shift further towards a lower token count.
> >
> > (c) **Visual encoder free VLMs (Chameleon):** While we do not have resources to train visual encoder-free VLMs, where the LLM is trained from scratch on a common input modality of both text and vision space, as mentioned above we do not foresee any reason for our scaling insights to not hold here. This is because the core reason for our key insights remains valid here as well — the LLM has to process a huge number of visual tokens that can be compressed. We also note that it is not straightforward how token compression can be performed in these types of architectures where a discrete tokenizer encodes the image.
> >
> > > if they [scaling laws] would hold as new more complex models are developed in the future.
> >
> > Based on our results, we believe that our scaling laws will hold for VLMs that have the structure outlined in our paper: a vision encoder, visual token projector, and LLM: one of the most common layouts for current VLMs. As compression algorithms get better and can retain more information in fewer tokens, the compute optimal point will most likely shift further towards using fewer tokens (e.g., the way compute optimal tokens shifted from 36 in LLaVA-PruMerge to only 1 in TokenPacker.) We hope that our scaling laws can help guide the future design of better and compute optimal VLMs.
> >
> > > proposed prompt-based token compression method adds complexity to the VLM pipeline
> >
> > While we do admit that our proposed query-based token compression may increase complexity within a VLM pipeline; we believe its complexity is not significantly greater than alternative methods, e.g., Llama-VID [5] which uses an entirely separate text encoder for prompt-based compression. In addition, in situations where the prompt is fixed, the query injection can be fully preprocessed and cached.
> >
> > Finally, we also would like to highlight that **query-based token compression is just meant to be the first step towards extreme token compression**. We simply wanted to highlight the importance of selecting only relevant tokens under extreme compression, motivated by our core contributions in the scaling laws.
> > > The paper does not discuss the training costs associated with the proposed compression method. It's possible that training such a method to be effective, especially at high compression ratios, could require significant computational resources.
> >
> > Token compression (especially extreme) **reduces the GPU-hours required to train** the projector module of VLMs and end-to-end VLM. This is because the number of tokens now passed as input to the LLM are 2 orders of magnitude smaller (e.g., 1 or 16) compared to 576, which greatly speeds up the training process (we saw up to 60% reduction in pretraining time and 40% reduction in instruction finetuning time on 4 A100’s). Note that the token compression modules are usually quite lightweight compared to LLM.
> >
> > > What is the inference time and performance trade-off?
> >
> > The exact variation with inference time is hard to capture, as it depends on a number of factors that vary depending upon the actual inference time device - the number of cores, parallelizable threads, etc. That is why, FLOPs (floating point operations per second) is the commonly used metric to compare the efficiency of two algorithms.
> >
> > We again thank the reviewer for asking critical and insightful questions and for their time spent reviewing our manuscript.  We hope that our response addresses their key concerns. Please let us know if you have any additional concerns!
> >
> > [1] [2405.09818] Chameleon: Mixed-Modal Early-Fusion Foundation Models
> >
> > [2] [2203.15556] Training Compute-Optimal Large Language Models
> >
> > [3] [2001.08361] Scaling Laws for Neural Language Models
> >
> > [4] [2407.02392] TokenPacker: Efficient Visual Projector for Multimodal LLM
> >
> > [5] [2311.17043] LLaMA-VID: An Image is Worth 2 Tokens in Large Language Models

---

> > > ### Author Response · Authors · 2024-11-22
> > > **Thank you for the feedback**
> > >
> > > Dear Reviewer 78vE,
> > >
> > > In our response above, we have tried to address all your comments. To summarize, you had concerns mainly around generalization of the scaling laws to different VLM architectures. Based on your feedback, we **added new results** on another VLM architecture that showed **similar trends as the original scaling laws** in our manuscript. We have also explained why the scaling laws can be expected to hold for most general VLM architectures, as they all operate on a similar principle.
> > >
> > > We thank you again for your feedback and time spent reviewing our work, which have helped strengthen our paper, and hope to hear back from you soon. We are also more than happy to address any additional concerns that you might have.

---

> ### Author Response · Authors · 2024-11-25
> **Hoping to hear back soon**
>
> Dear Reviewer 78vE,
>
> As the end of the discussion phase is approaching, we were wondering if you had any additional questions or concerns.
>
> To summarize, in our response above, we have addressed all your concerns mainly around **generalization of our scaling laws** to different architectures. We added a discussion of why it will generalize and also **added new results** that empirically validate the same by showing performance on other VLM architectures followed similar scaling trends as in our original manuscript.
>
> We appreciate your time and valuable feedback which has improved our work, and we hope to hear back from you soon.

---

> ### Author Response · Authors · 2024-12-01
> **Looking forward to your response**
>
> Dear Reviewer 78vE,
>
> As the end of the discussion phase is approaching, we look forward to hearing back from you regarding our response. We hope our shared and reviewer responses have been able to address your questions, and we are more than happy to address any lingering questions or concerns.

---

> ### Author Response · Authors · 2024-12-02
> **Looking forward to your response**
>
> Dear Reviewer 78vE,
>
> As today is the last day for the discussion phase, we humbly request you to review our response to your concerns and comments and hope that it allows you to assess the work more positively.

---

> > ### Author Response · Authors · 2024-12-03
> > **Looking forward to your response**
> >
> > Dear Reviewer 78vE,
> >
> > As today is the last day for the discussion phase, we humbly request you to review our response to your concerns and comments and hope that it allows you to assess the work more positively.

---

### Author Response · Authors · 2024-11-19
**General Response (1/2)**

We thank all the reviewers for their time spent reviewing our manuscript and providing thoughtful comments and feedback. We are glad that the reviewers found the core contribution of our paper, i.e., the scaling laws for VLMs, as “interesting” and “providing a deeper understanding of VLM optimization” (R78vE) and “a novel and valuable insight into VLM efficiency” (RWUh0, Rjj75, RiUzh), supported by “comprehensive and well designed experiments” (RLaf9, RWUh0, R78vE).

Most of the concerns raised were around ablations and comparisons for query-based token compression, which we have addressed in this shared response. We would also like to point out that the main contribution of our paper is its VLM scaling laws and not the query-based compression.

Building off reviewer feedback, we have also uploaded a new version of the manuscript with key changes highlighted in blue.


## Core Contribution
Our work mainly focuses on understanding and characterizing the tradeoff between tokens and parameters while optimizing for VLM efficiency, via **scaling laws**. Our scaling laws **led to multiple takeaways and valuable insights** (as acknowledged by all the reviewers), with the key insight being extreme token compression is required for compute optimal inference for visual reasoning and understanding tasks.

Based on the above insights, we wanted to highlight the importance of incorporating query in extreme token compression algorithms to only retain relevant information. We tried to convey this message by building over existing state-of-the-art approaches like TokenPacker, which does not incorporate query. This is simply a small section of the manuscript, aimed to empirically validate the importance of query-based compression.
## Additional Compression Comparisons with Llama-Vid and VoCo-LLaMA
One of the concerns raised was about the comparison of query-based compression in our work with LLaMA-VID and VoCo-LLaMA.

**LLaMA-Vid:** First, there are key design differences with LLaMA-Vid. They use only a single “context” token that is supposed to capture information based on user query, while all other “content” tokens are compressed independent of query. In contrast, in our work, we use query for all compressed tokens. More importantly, coming back to our main goal of understanding the significance of the query, we note that Llama-Vid ablates the query’s importance by removing the “context” token altogether. However, this makes it unclear whether the drop in performance was due to a reduction in tokens (from 2 to 1) or a lack of incorporating query in token compression. Thus, we decided to empirically validate the importance of query, by building over existing algorithms like TokenPacker.

Nevertheless, in the table below, we compare the performance of our method to LLaMA-VID’s *reported* performance at similar token compression levels and show that we are able to outperform it in certain tasks **despite LLaMA-VID utilizing a stronger vision encoder** [2]. Both approaches are competitive, which also validates the key point we wanted to make that query-based compression is necessary under extreme compression. We would also like to note that architecturally, LLaMA-VID uses a separate text decoder model to process the user query, while our method utilizes the existing LLM within the VLM model.

| Token Count | Model       | GQA   | POPE  | SQA   | TextVQA |
|-------------|-------------|-------|-------|-------|---------|
| 16         | LLaMA-VID   | 58.2  | 83.1  | 67.4  | 50.8    |
|             | QueCC       | 59.0  | 83.4  | 70.7  | 51.3    |
| | | | | | |
| 4           | LLaMA-VID   | 56.2  | 83.5  | 68.7  | 49.1    |
|             | QueCC       | 56.5  | 81.8  | 68.6  | 48.7    |


**VoCo-LLaMA:** Reviewers WUho and jj7r also raised questions about comparisons with VoCo-LLaMA [3]. Their approach, although highly performant in terms of accuracy, *requires processing all the 576 visual tokens through the LLM*, to get a compressed token representation. This compressed representation is then cached and can be used when running the inference on an image for the second time. However, due to the cost of getting the cached representation, this *does not give any notable inference efficiency gains* under the commonly studied setting of varying images (e.g. monitoring systems, autonomous driving) considered in our work and other recent approaches [4,5]. In these works (including ours), tokens are compressed to say just 1,4, or 16 tokens using a lightweight module, before passing through the LLM (in contrast, VoCo-LLaMA processes all 500+ visual tokens in the LLM). This significantly reduces the inference cost for any image (and is not constrained to the setting of repeated inference on the same image). Thus, we do not believe that VoCo-LLaMA is a fair comparison baseline for our work.

---

> ### Author Response · Authors · 2024-11-19
> **General Response (2/2)**
>
> ## Ablations for Token Compression Algorithm:
> Reviewers WUho and jj75 also requested ablations of our query-based compression that shows the impact of (a) user query information injection, and (b) the convolutional downsampling on performance. We report the ablations for both components below.
>
> Based on the results shown below, it can be seen at extreme levels of compression that **combining query and convolution can magnify the benefits** of either adding only query or only convolution, e.g., TextVQA performance at token count one increased by 0.7 percentage points (pp) with both convolution and query while using only one of the components led to at most 0.2 pp increase. In addition, combining the two can **mitigate performance drops** that are associated with utilizing only query or convolution, as seen in MMB at one token where using only convolution drops performance by more than 1 pp but performance can not only be restored but also improved when adding query, eventually outperforming the baseline by 0.7 pp; a similar situation can be seen for MME.
> | Token Count | Model               | GQA   | MMB   | MME     | POPE  | SQA   | TextVQA | VizWiz | VQAv2 |
> |-------------|---------------------|-------|-------|---------|-------|-------|---------|--------|-------|
> | 1           | Conv and Query      | 53.5  | **59.4**  | **1269.1**  | **81.3**  | **69.9**  | **46.9**    | 44.1   | **67.3**  |
> |             | Query Only          | 53.3  | 59.2  | 1267.7  | **81.3**  | 68.8  | 46.3    | 41.7   | 66.6  |
> |             | Conv Only           | **53.6**  | 57.5  | 1215.5  | 80.6  | 69.1  | 46.4    | **45.6**   | 66.7  |
> |             | No Conv, No Query   | 53.4  | 58.7  | 1262.4  | 80.7  | 69.4  | 46.2    | 41.1   | 66.9  |
> | | | | | | | | | |
> | 4           | Conv and Query      | 56.5  | **62.1**  | **1390.3**  | 81.8  | 68.6  | 48.7    | 45.0   | **70.6**  |
> |             | Query Only          | 56.4  | 62.0  | 1345.9  | **82.3**  | **70.7**  | 48.8    | **46.5**   | **70.6**  |
> |             | Conv Only           | **56.7**  | 60.6  | 1310.4  | 82.1  | 69.0  | **49.4**    | 41.3   | 70.5  |
> |             | No Conv, No Query   | 56.2  | 61.5  | 1347.6  | 81.7  | 68.5  | 49.2    | 45.7   | 70.5  |
> | | | | | | | | | |
> | 16          | Conv and Query      | **59.0**  | 62.2  | **1408.0**  | 83.4  | **70.7**  | 51.3    | 47.7   | **74.5**  |
> |             | Query Only          | 56.6  | 61.4 | 1354.3  | 82.1 | 69.6 | 50.7   | 41.2   | 71.5 |
> |             | Conv Only           | 58.9  | 62.5 | 1402.3  | 82.5  | 69.6 | **52.6**    | 45.7   | 74.1  |
> |             | No Conv, No Query   | 58.9  | **62.7**  | 1378.8  | **83.7**  | 68.1  | 52.5    | **50.5**   | 74.4  |
>
>
>
> [1] [2311.17043] LLaMA-VID: An Image is Worth 2 Tokens in Large Language Models
>
> [2] [2303.15389] EVA-CLIP: Improved Training Techniques for CLIP at Scale
>
> [3] [2406.12275] VoCo-LLaMA: Towards Vision Compression with Large Language Models
>
> [4] [2403.15388] LLaVA-PruMerge: Adaptive Token Reduction for Efficient Large Multimodal Models
>
> [5] [2407.02392] TokenPacker: Efficient Visual Projector for Multimodal LLM

---

### Meta-Review · Area_Chair_RdiG · 2024-12-22

**Metareview:**

The manuscript studies inference time optimization by means of balancing the LLM size with respect to the number of visual tokens, which can lead to a favorable trade-off in visual reasoning tasks. The proposed token compression method seems to outperform existing techniques, especially under extreme token reduction settings. Concerns were raised about the generalizability of the scaling laws, and it was pointed out (also by the authors) that there are families of tasks which target fine-grained visual comprehension whose performance will suffer in the extreme visual token compression schemes. Reviewers also pointed out the lack of comparisons with established models like LLaMA-VID and VoCo-LLaMA, as well as the absence of ablation studies to identify which components drive the method's success.

After the discussion phase the work remains borderline, but I'm going to recommend acceptance as this issue deserves more attention in the VLM community. I urge the authors to incorporate the feedback from the discussion phase.

**Additional Comments On Reviewer Discussion:**

Several ablations were requested and ultimately provided by the authors.

---

### Decision · Program_Chairs · 2025-01-22

Accept (Poster)